# Rapid transgenerational adaptation in response to intercropping reduces competition

Laura Stefan[1,2]*, Nadine Engbersen[1], Christian Schöb[1,3]

[1]Institute of Agricultural Sciences, ETH Zurich, Zurich, Switzerland; [2]Plant Production Systems, Agroscope, Nyon, Switzerland; [3]Área de Biodiversidad y Conservación, Universidad Rey Juan Carlos, Móstoles, Spain

**Abstract** By capitalising on positive biodiversity–productivity relationships, intercropping provides opportunities to improve agricultural sustainability. Intercropping is generally implemented using commercial seeds that were bred for maximal productivity in monocultures, thereby ignoring the ability of plants to adapt over generations to the surrounding neighbourhood, notably through increased complementarity, that is reduced competition or increased facilitation. This is why using monoculture-adapted seeds for intercropping might limit the benefits of crop diversity on yield. However, the adaptation potential of crops and the corresponding changes in complementarity have not been explored in annual crop systems. Here we show that plant–plant interactions among annual crops shifted towards reduced competition and/or increased facilitation when the plants were growing in the same community type as their parents did in the previous two generations. Total yield did not respond to this common coexistence history, but in fertilized conditions, we observed increased overyielding in mixtures with a common coexistence history. Surprisingly, we observed character convergence between species sharing the same coexistence history for two generations, in monocultures but also in mixtures: the six crop species tested converged towards taller pheno-types with lower leaf dry matter content. This study provides the first empirical evidence for the potential of parental diversity affecting plant–plant interactions, species complementarity and there-fore potentially ecosystem functioning of the following generations in annual cropping systems. Although further studies are required to assess the context–dependence of these results, our find-ings may still have important implications for diversified agriculture as they illustrate the potential of targeted cultivars to increase complementarity of species in intercropping, which could be achieved through specific breeding for mixtures.

*For correspondence:
laura.stefan@m4x.org

**Competing interest:** The authors declare that no competing interests exist.

## Editor's evaluation

This study reports that interactions between crop species grown over three generations in mixture instead of monoculture become less competitive/more facilitative, suggesting a way to breed for increased mixture yield. This fundamental finding is of high interest to the fields of ecology and agriculture, as well as society seeking new solutions to satisfy increasing food demands. The meth-odological approach is compelling, yet distinguishing between reduced competition and increased facilitation remains challenging.

## Introduction

Following decades of studies demonstrating the positive relationship between species diversity and plant primary productivity in natural systems (*Spehn et al., 2005*; *Tilman et al., 2001*), intercropping,

**eLife digest** Plants have two ways of interacting with each other: they can compete with each other if they use the same resources; or they can 'help' each other in what is known as facilitation, for example, when a larger plant protects a smaller plant in harsh environments. These interactions can vary over several generations in response to changes in the environment or the surrounding plant community. For instance, in plant communities formed by many different species, like in most natural systems, competition usually decreases over time as the plants 'learn' to grow together.

In agriculture, intercropping – defined as growing at least two species of crop at the same time on the same field – takes advantage from a reduction in competition. The idea is that planting two species that grow differently together will lead to less competition than having a single crop because the two species will use slightly different resources, or use them at different times. However, intercropping has traditionally overlooked changes in the interactions between plants as a result of the crop species evolving after being grown together for generations. Indeed, farmers that practice intercropping generally use standard seeds that have been bred to produce high yields when planted on their own, in what is known as monoculture. If plants can adapt and become less competitive when they are grown together over several generations, then using these standard seeds might limit the success of intercropping.

Stefan, Engbersen and Schöb wanted to know whether crop species adapt to the levels of plant diversity surrounding them over generations, and if so, how they do it. To find this out, they investigated how competition and facilitation changed when six crop species (wheat, oat, lentil, coriander, flax and camelina) that grow annually were grown together in different combinations over several generations. Stefan, Engbersen and Schöb started off with seeds normally used for growing these crops on their own, and planted them either on their own, or in different combinations of two or four species. They then repeated the experiment over the course of three years, each year using seeds from the previous year, recording both crop yields and changes in how the plants interacted with each other.

The experiments showed that interactions among these annual crops shifted towards reduced competition and/or increased facilitation when the plants were growing alongside the same crops as their parents did in the previous two generations.

Improving and promoting the development of intercropping is essential for agricultural sustainability, as it could offer alternatives to intensive monocultures (crops grown on their own that require increased resources). Stefan, Engbersen and Schöb's findings are relevant for programmes aimed at developing seeds for intercropping, as they highlight the importance of including diversity when developing these seeds. However, before these results can be used in the field, longer experiments (of more than three generations) in different environments should be carried out to confirm the findings. Another question that remains open is what the mechanisms underlying adaptations to intercropping are: more in-depth research will be needed to determine whether the changes observed have a genetic basis.

that is growing more than two species in the same field during the same period, has been increasingly considered as a promising option to increase agricultural sustainability (*Gurr et al., 2016*; *Brooker et al., 2015*; *Vandermeer, 1992*). The productivity benefits of increasing species diversity rely on two main mechanisms, namely selection effects and complementarity effects, the latter encompassing both facilitation and niche differentiation (*Loreau and Hector, 2001*; *Hooper et al., 2005*). In perennial natural grasslands, complementarity effects have been shown to increase over time due to evolutionary processes (*Zuppinger-Dingley et al., 2014*; *van Moorsel et al., 2019*; *van Moorsel et al., 2019*). Notably, greater species complementarity can result from evolutionary changes (*Anderson et al., 2011*) ▬ that is changes in gene frequency ▬ or from heritable epigenetic changes (*Verhoeven et al., 2016*) affecting species traits in response to surrounding plant diversity, which either increases niche differentiation (i.e. reduces competition) or increases facilitation (*Schöb et al., 2018*; *Meilhac et al., 2020*).

The mechanisms of selection for plant facilitation remain very poorly understood (*Bronstein, 2009*; *Brooker, 2008*). Whereas some facilitative traits characterizing facilitator species are well known in

some systems — for example legume species fostering soil nitrogen enrichment (*Wright et al., 2017*), microclimate amelioration through shading by large canopy (*Aguirre et al., 2021*), nectar reward to attract pollinators (*Losapio et al., 2021*) — traits of facilitated species remain much more obscure (*Bronstein, 2009*). These 'facilitated' traits, which allow organisms to benefit from their neighbours, may be a target of evolutionary selection in natural systems (*Bronstein, 2009*), and it is therefore reasonable to think that they might depend on neighbour identity (*Schöb et al., 2018*). The neighbour-dependent evolution of facilitation in grassland plant communities was demonstrated by *Schöb et al., 2018*, who showed that selection for net facilitative interactions was favoured in plant mixtures.

The evolutionary potential of plant–plant interactions in diverse communities has tremendous implications for the diversification of agricultural systems (*Isbell et al., 2017*). This is of particular relevance for mixed cropping systems, where the use of commercial seeds domesticated and bred for maximum yield in monoculture is the norm (*Thrall, 2012*). These commercial varieties have been selected to express a particular phenotype or traits that would lead to the best yield in monoculture. Yet the optimal monoculture phenotype might not necessarily be the most adequate to promote positive diversity effects in mixtures, and may actually compromise the diversity benefits (*Thrall, 2012*; *Chen et al., 2021*; *Chacón-Labella et al., 2019*; *Wuest et al., 2021*; *Annicchiarico, 2019*). Despite the paramount importance of this question, the yield potential of mixture-adapted varieties is, to our knowledge, unknown, as are the trait differences of monoculture- compared to mixture-adapted crops.

Furthermore, the evolution and occurrence of plant–plant interactions is notably context-dependent (*Bertness and Callaway, 1994*). This environmental dependence of the direction and strength of plant-plant interactions has been conceptualised as the stress–gradient hypothesis (SGH), which suggests that competition between plants is stronger and more important in benign environments — where resources are abundant — while facilitation is more likely to occur in harsher environments — where resources are scarce (*Soliveres et al., 2015*; *Maestre and Cortina, 2004*; *Maestre et al., 2009*). In the context of intercropping, this means that competition would be the dominant interaction in highly productive systems, and therefore, the benefits of increasing niche differentiation — thereby reducing competition — would be higher (*Li et al., 2020*; *Stefan et al., 2021a*). On the other side of the stress gradient, enhanced facilitation in resource-poor, low-productive systems may also increase diversity effects (*Soliveres et al., 2015*). Therefore, the effect of environmental severity on plant–plant interactions and biodiversity effects in intercropped systems is unclear.

In this project, we determined whether, how, and under which soil fertility conditions crop species adapt over three generations to the level of plant diversity that they are surrounded by. We investigated how plant–plant interactions, that is competition and facilitation, and plant traits changed within different coexistence histories over time, and whether these changes translated into yield benefits. To that end, we conducted an intercropping experiment in Switzerland with six different crop species belonging to four functionally different phylogenetic groups, namely wheat, oat, lentil, flax, camelina, and coriander. These species are commonly cultivated in Europe as monocrops and some of them — for example oat, lentil, camelina — are also partly cultivated in intercrops (*Neumann et al., 2007*; *Kraska et al., 2004*). We used commercially available seeds commonly used for monoculture practices and selected, whenever possible, open-pollinated varieties as seed source to provide the genetic variability needed for evolutionary processes to occur. The mesocosms — square plots of 0.25 m² — included monocultures, 13 different 2-species mixtures, four different 4-species mixtures, and isolated single plants, and was replicated in two different fertilizing conditions. To assess potential transgenerational changes, we repeated the experiment over the course of three years with seeds from plants grown from either monocultures, mixtures, or single individual plants of the previous year (*Figure 1*, *Figure 1—figure supplement 1*). In the third year, we assessed plant–plant interactions within each community using the Relative Interaction Index (*Armas et al., 2004*), a symmetrical and standardized index that compares the performance in terms of grain yield of a plant growing in a community to its performance when growing in isolation (*Michalet et al., 2014*; *Schöb et al., 2014*) (see Methods). This index takes a plant's eye view by quantifying plant–plant interaction intensity experienced by crop species in monocultures and mixtures. We therefore directly quantify species complementarity underlying classical biodiversity effects (*Armas et al., 2004*; *Schöb et al., 2018*). Yield was also used to derive the metrics of biodiversity effects following the method of *Loreau and Hector, 2001*. Finally, we measured standard above-ground plant traits — that is plant height, plant width, Specific Leaf

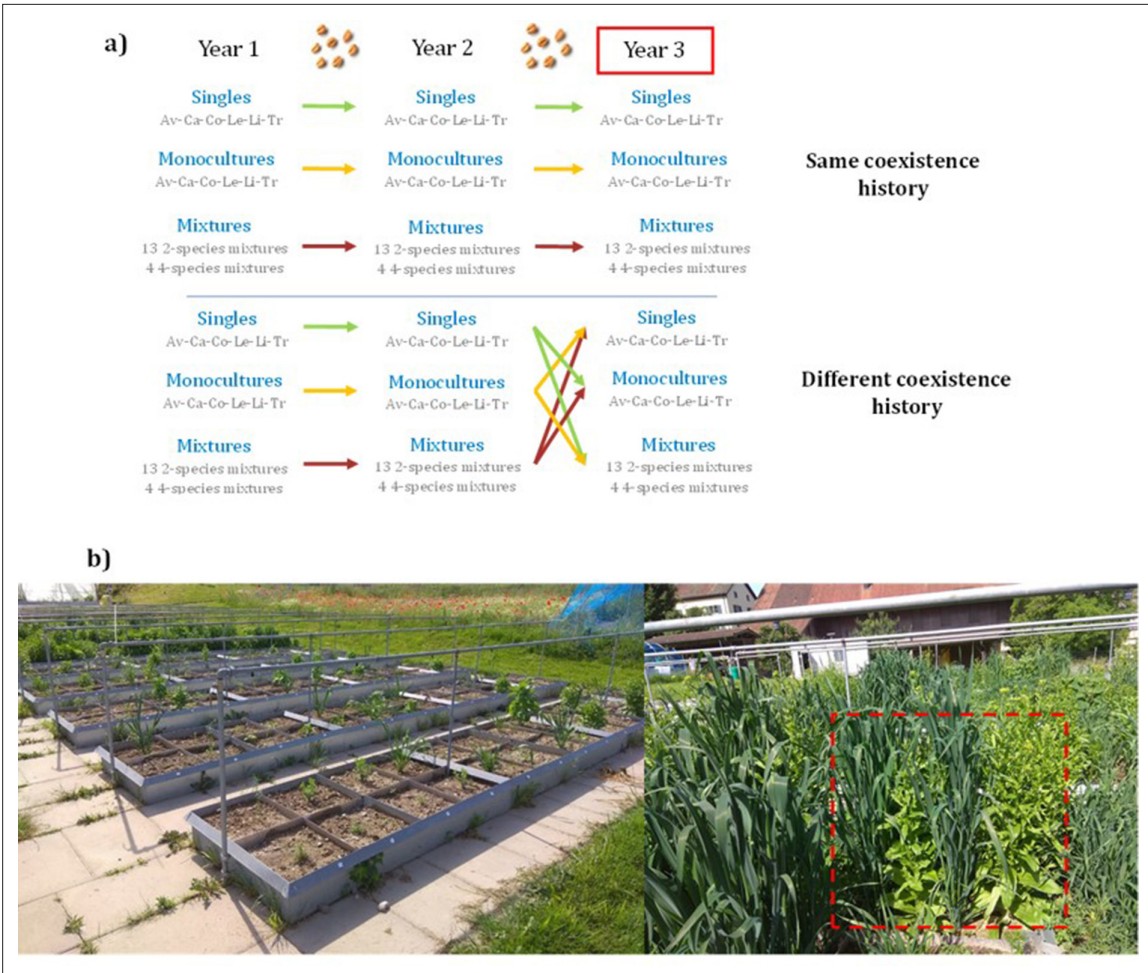

**Figure 1.** Experimental design. (**a**) Six crop species were used to sow single plant individuals (*Loreau and Hector, 2001*), monocultures (*Loreau and Hector, 2001*), 2-species mixtures (*Schöb et al., 2018*) and 4-species mixtures (*Brooker et al., 2015*) in 2018 (Year 1) (see *Supplementary file 1s* for the list of species mixtures); seeds were collected at the end of the growing season and resown in 2019 (Year 2) in the same diversity setting as their previous generation. Seeds were collected again and resown in 2020 (Year 3), this time either in the same community their seeds were collected from [same coexistence history], or in a community different to the one of their parents [different coexistence history] (n=468 plots). This process was replicated in two different fertilizing conditions. We expected that crops growing in the same community as their parents would have adapted over the two generations, and therefore would exhibit less competition and have higher productivity than crops growing in a community different to the one of their parents. Av: *Avena sativa*; Ca: *Camelina sativa*; Co: *Coriandrum sativum*; Le: *Lens culinaris*; Li: *Linum usitatissimum*; Tri: *Triticum aestivum* (b) Left: part of the experimental garden, showing the plots within beds, and planted with single individuals. Right: a plot is outlined in red, showing a 2-species mixture, with oat alternated with camelina.

The online version of this article includes the following figure supplement(s) for figure 1:

**Figure supplement 1.** Pictures of the experimental plots.

Area (SLA), Leaf Dry Matter Content (LDMC), and mass per seed — and investigated whether there was a change in mean and variability at the species and community levels in response to the coexistence history of the community.

We hypothesized that crop mixtures composed of offspring of plants that had been grown for two generations in the same community type (as their offspring do now) would show increased niche differentiation (i.e. less competition) and/or increased facilitation compared to communities of offspring of plants that had been grown in a different community type (as their offspring do now). We also expected that these changes in plant–plant interactions would lead to changes in complementarity effects in crop mixtures, that is communities with the same coexistence history would show higher complementarity effects than the communities with a different history. We hypothesised that the increased niche differentiation would be due to enhanced character displacement. Finally, following

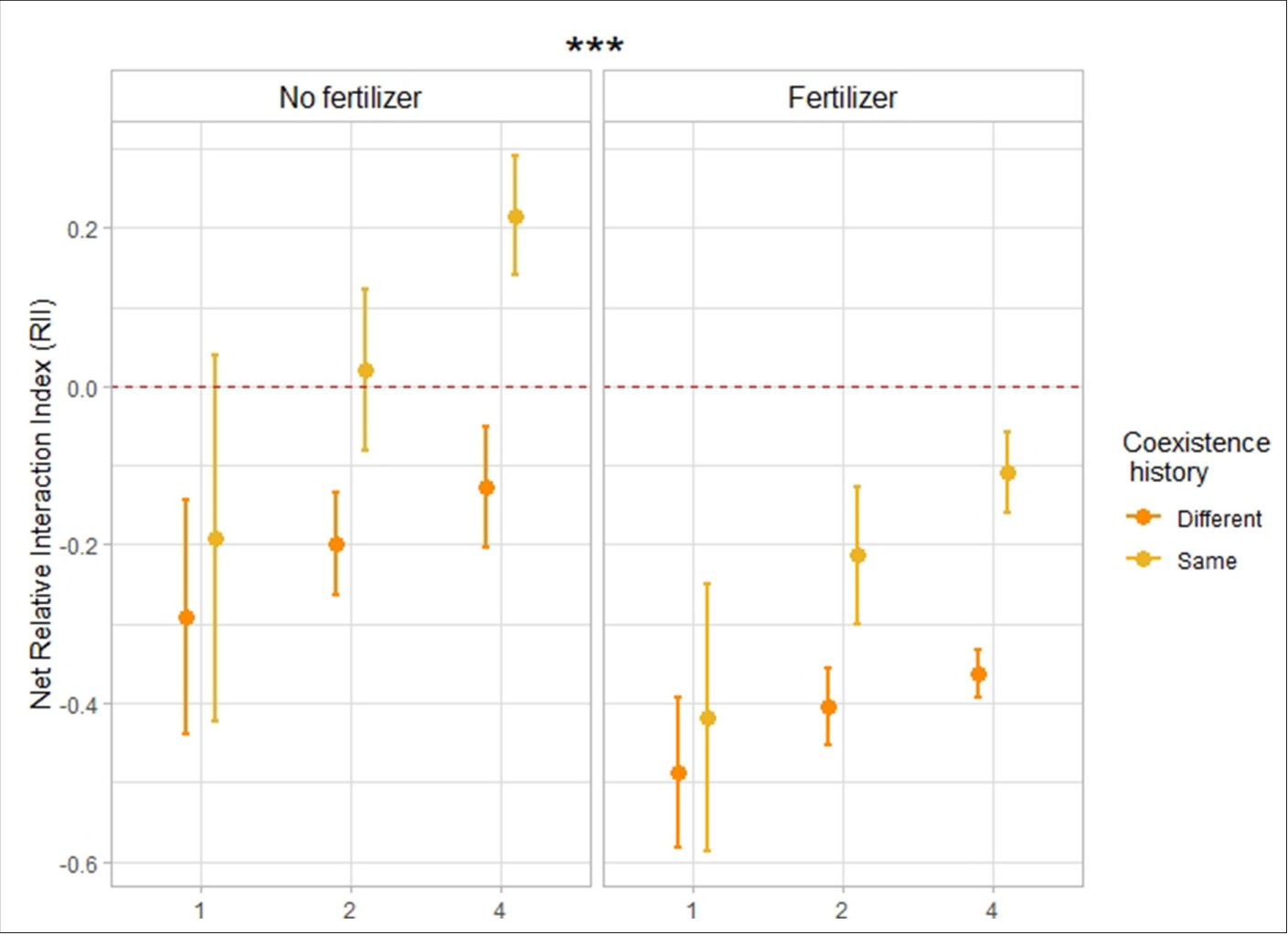

**Figure 2.** Relative Interaction Index in response to coexistence history and fertilization. Net relative interaction index of monocultures, 2- and 4-species mixtures in response to coexistence history, for fertilized and unfertilized conditions. n=276. Dots represent the mean values across plots; lines represent the standard error. Stars placed above or next to the results represent the significance of the coexistence history effect. The net Relative Interaction Index (RII) compares the performance of plants growing in communities to the performance of single plants growing alone, with the same coexistence history treatment as the focal plant (see Methods). Negative RII indicates competition within a community, positive RII indicates facilitation. The closer this index gets to 1, respectively –1, the stronger the facilitation, respectively competition. 'Same coexistence history' indicates that crops were grown in the same community type as their parents (i.e. monocultures with seeds coming from monocultures, 2-species mixtures with seeds coming from the same 2-species mixtures [e.g. oat-lentil with seeds coming from oat-lentil], 4-species mixtures with seeds coming from the same 4-species mixtures [e.g. oat-lentil-coriander-flax with seeds coming from oat-lentil-coriander-flax]). "Different coexistence history" refers to crops grown in a community type different to the one of their parents (i.e. monocultures with seeds coming from singles, monocultures with seeds coming from mixtures, mixtures with seeds coming from singles, mixtures with seeds coming from monocultures). See *Supplementary file 1a* for the complete statistical analysis, and *Figure 2—figure supplement 1* for the corresponding boxplots.

The online version of this article includes the following figure supplement(s) for figure 2:

**Figure supplement 1.** Effects of coexistence history and crop species number on net Relative Interaction Index (RII), for fertilized and unfertilized conditions.

the stress gradient hypothesis (*Bertness and Callaway, 1994*), we expected more facilitation and/or less competition in conditions of reduced soil fertility.

## Results and discussion

Results from the third year showed that plant–plant interactions shifted towards increased complementarity, that is weaker competition and/or stronger facilitation — distinguishing between these two

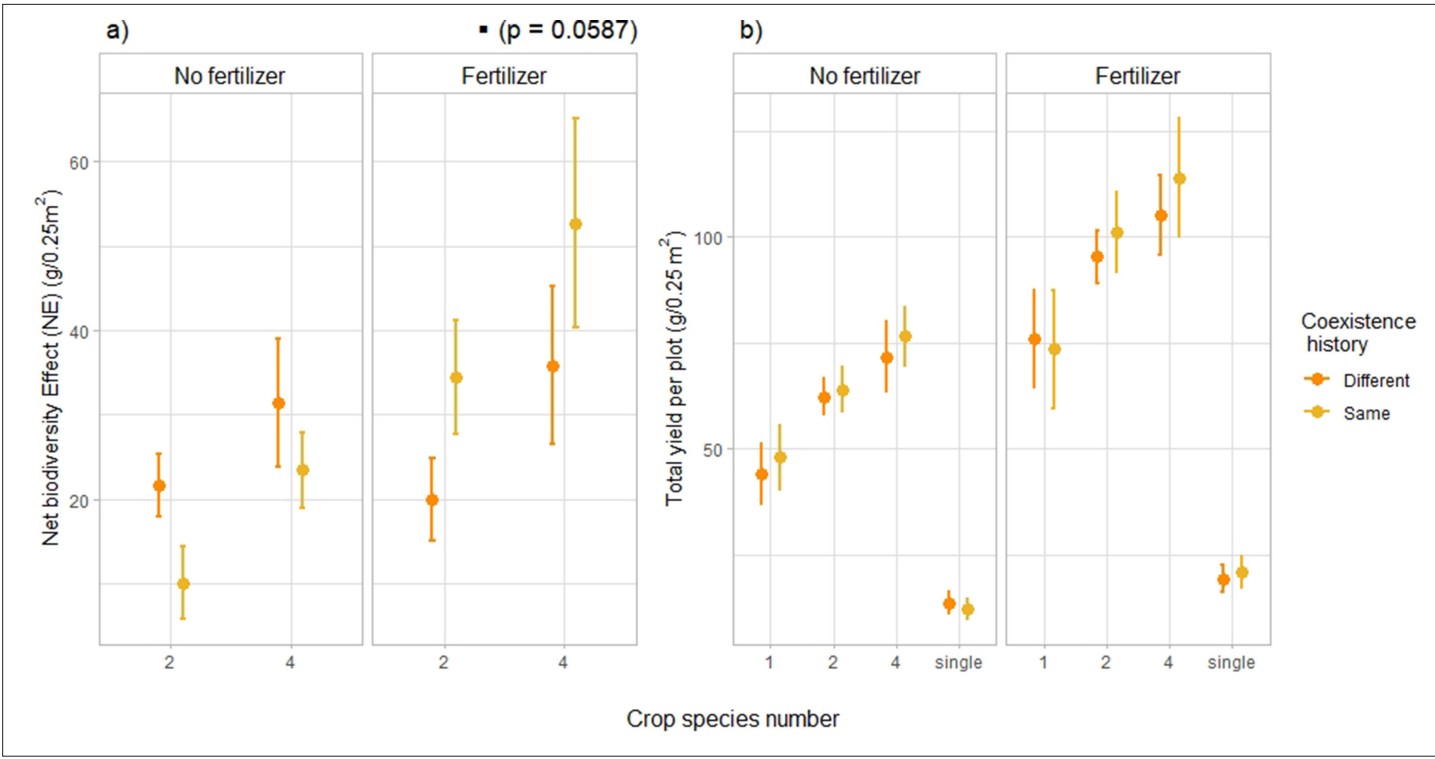

**Figure 3.** Effects of coexistence history on net biodiversity effects (**a**) and total yield per plot (**b**). Effects of coexistence history and crop species number on (**a**) net biodiversity effect – reflecting the yield advantage of mixtures compared to monocultures – and (**b**) total yield per plot in fertilized and unfertilized plots. (**a**) n=204; (**b**) n=276. Dots represent the mean values across plots; lines represent the standard error. Stars or dots placed above or next to the legend represent the significance of the coexistence history effect. 'Same coexistence history' indicates that crops were grown in the community their seeds were collected from. 'Different coexistence history' refers to crops grown in a community different to the one of their parents. See SI *Supplementary file 1c-f* for the complete statistical analysis, *Figure 3—figure supplement 1* for complementarity and selection effects, and *Figure 3—figure supplements 2 and 3* for the corresponding boxplots.

The online version of this article includes the following figure supplement(s) for figure 3:

**Figure supplement 1.** Effects of coexistence history and crop species number on net biodiversity effect.

**Figure supplement 2.** Effects of coexistence history and crop species number on complementarity effect (**a**) and selection effect (**b**) in fertilized and unfertilized plots.

**Figure supplement 3.** Effects of coexistence history and crop species number on total yield per plot.

**Figure supplement 4.** Effects of coexistence history of total yield per plot, per species combination.

mechanisms was not possible in this study — when the plants were growing in the same community types as their two previous generations (*Figure 2*, *Figure 2—figure supplement 1*, *Supplementary file 1a*). More precisely, the net Relative Interaction Index, which compares the performance of focal plants growing in communities to the performance of single plants growing alone — focal plants and single plants having the same coexistence history treatment — was significantly higher (+54% [*F*=30.4; p-value < 0.001; n=276]) when the offspring was grown in the same community type as their parents did than when they were growing in a community type different to the one of their parents (*Figure 2*). Pairwise comparisons further showed that this effect of coexistence history was particularly true in mixtures and only a trend in monocultures, for both fertilizing conditions (*Supplementary file 1b*). This notably demonstrates that in mixtures, mixture-adapted communities (i.e. with the same coexistence history) exhibited less competition and/or more facilitation than monoculture-adapted communities or single-adapted communities (i.e. with a different coexistence history). Furthermore, when looking at the effect of fertilization, we observed that competition was weaker and/or facilitation was stronger in unfertilized plots (*Figure 2*, *Figure 2—figure supplement 1*;+64% *F*=44.5; p-value < 0.001; n=276), which is in accordance with the stress–gradient hypothesis.

This shift in plant–plant interactions was accompanied by a similar shift in net biodiversity effect (NE) in fertilized plots (*Figure 3a*, *Figure 3—figure supplement 1a*). Net biodiversity effect — or

overyielding — represents the deviation from the expected yield in the mixture, based on the yield of the corresponding monocultures with the same coexistence history as the focal mixture (*Loreau and Hector, 2001*). The interaction between fertilization and coexistence history had a significant effect on NE (*Supplementary file 1c*, [$F=9.60$, p-value = 0.0023, n=204]). Posthoc pairwise comparisons further showed that under fertilized conditions, across all species combinations, NE was on average 58% higher with the same coexistence history than with a different coexistence history (*Figure 3a*, *Supplementary file 1d* p-value of the pairwise comparison: 0.0587). This indicates that in fertilized plots, overyielding of crop mixtures tended to be higher with mixture-adapted individuals compared to monoculture-adapted and single-adapted individuals. In unfertilized plots we did not observe the same result, which suggests that even though the shifts in plant–plant interactions were consistent across fertilizing conditions, overyielding was not. When looking at the partitioning of net effects into complementarity and selection effects (*Loreau and Hector, 2001*), we observed a significant interaction effect between fertilizer, coexistence history and planted diversity on selection effects (*Figure 3—figure supplement 2b*, *Supplementary file 1c*, [$F=4.09$, p-value = 0.045, n=204]). More precisely, for 4-species mixtures under fertilized conditions, SEs were higher in plant communities composed of offspring of plants that had been grown in the same community type (as their offspring do now) than plant communities of offspring of plants that had been grown in a different community type (+109%, *Figure 3—figure supplement 2b*, *Supplementary file 1c*, [p-value of the pairwise

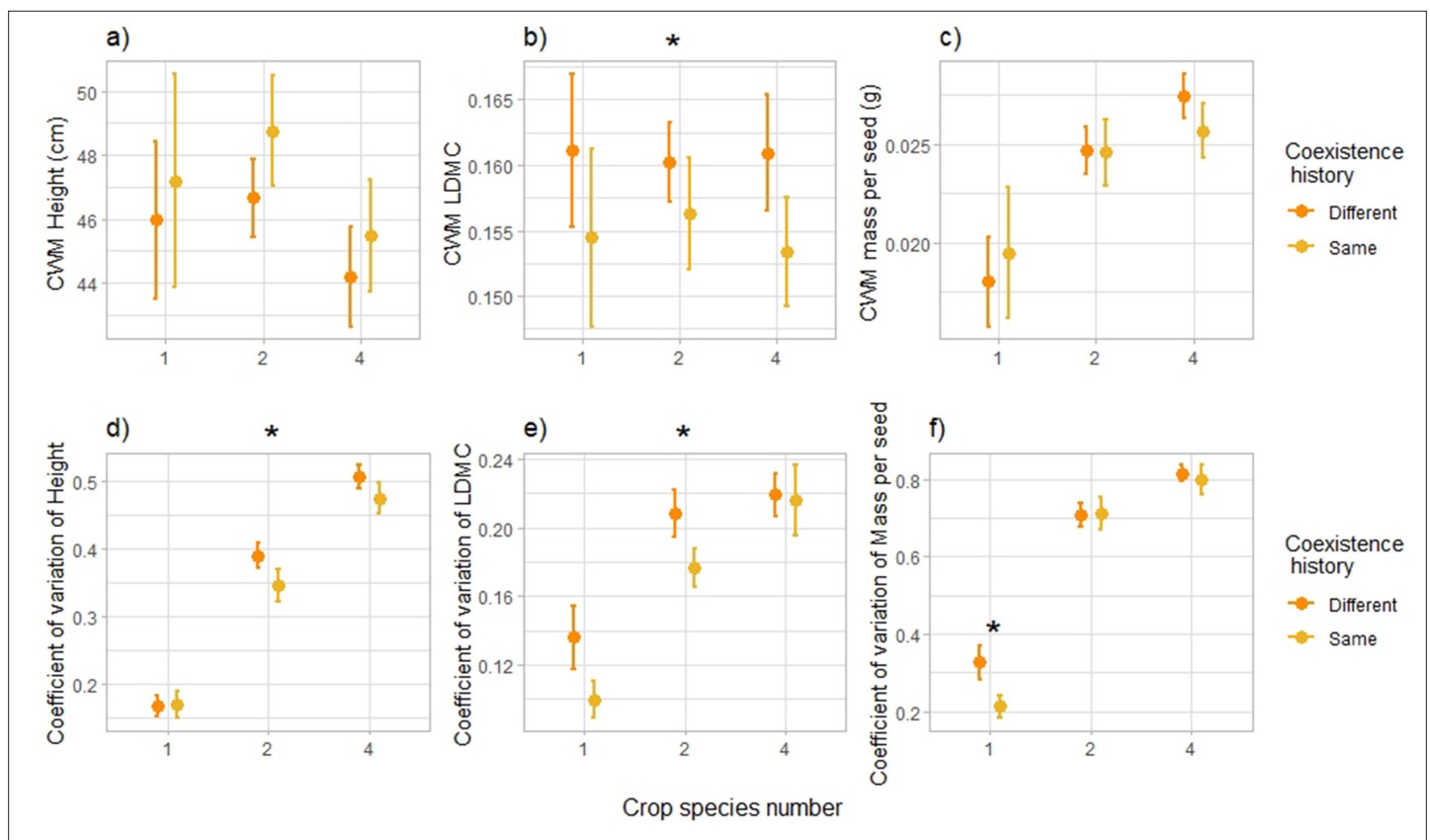

**Figure 4.** Community-level trait responses to coexistence history. Effects of coexistence history and crop species number on community-weighted mean (CWM) of height (in cm) (**a**), Leaf Dry Matter Content (LDMC) (**b**), and mass per seed (in g) (**c**), and on coefficient of variation at the community level of height (**d**), LDMC (**e**), and mass per seed (**f**). n=271. Dots represent the mean values across plots; lines represent the standard error. Stars placed above represent the significance of the coexistence history effect. See *Supplementary file 1l-p* for the complete statistical analyses, and Fig. S6-7 as well as *Supplementary file 1g-k* for responses at the species level.

The online version of this article includes the following figure supplement(s) for figure 4:

**Figure supplement 1.** Effects of coexistence history and crop species number on mean height (in cm) (**a**) and LDMC (**b**) Dots represent the averaged values across species and plots; lines represent the standard error.

**Figure supplement 2.** Mean height (cm) (**a**) and LDMC (**b**) according to their coexistence history, for the six species considered in our study.

comparison: 0.0286]). Coexistence history did not affect complementarity effects (*Figure 3—figure supplement 2a*, *Supplementary file 1c*, [$F$=1.57, p-value = 0.21, n=204]), nor total yield (*Figure 3b*, *Figure 3—figure supplement 3*, *Figure 3—figure supplement 4*, *Supplementary file 1f*, [$F$<1, p-value > 0.5, n=276]).

To investigate the ecological mechanisms behind the shift in plant–plant interactions with coexistence history, we assessed the response of standard above-ground plant traits and compared the average values and coefficients of variation at the species and community levels of single-, monoculture- and mixture-adapted varieties. Results pointed towards a reduction in trait variation at the community level, notably of height and leaf dry matter content (*Figure 4*): the coefficient of variation of height was lower in plant communities composed of offspring of plants that had been grown in the same community type (as their offspring do now) compared to plant communities of offspring of plants that had been grown in a different community type (–9%, *Figure 4d*, *Supplementary file 1l*, [$F$=3.93, p-value = 0.049, n=271]), and for leaf dry matter content it was 15% lower with the same history compared to a different history (*Figure 4e*, *Supplementary file 1o*, [$F$=4.18, p-value = 0.042, n=271]). Furthermore, the coefficient of variation of mass per seed was also lower under the same history compared to a different history, but this effect was only significant in monocultures (–33%, *Figure 4f*, *Supplementary file 1p*, [$F$=5.48, p-value = 0.020, n=271]). The community-weighted means of plant traits (CWM, calculated at the community level) further suggest that when growing in the same community type as their parents, plants seemed to converge towards taller individuals with lower leaf dry matter content. Indeed, the community-weighted mean of leaf dry matter content was significantly lower in plant communities composed of offspring of plants that had been grown in the same community type compared to plant communities of offspring of plants that had been grown in a different community type (–3%, *Figure 4b*, *Supplementary file 1o*, [$F$=4.33, $P$-value = 0.039, n=271]); height community-weighted mean was not significantly different between coexistence histories (*Figure 4a*, *Supplementary file 1l*, [$F$<1, p-value = 0.48, n=271]), but at the species level we did observe a consistent increase in plant height under the same coexistence history compared to a different history (*Figure 4—figure supplement 1*, *Figure 4—figure supplement 2*, *Supplementary file 1g*, [$F$=4.29, p-value = 0.040, n=1,726]). We observed similar consistent responses of leaf dry matter content at the species level (*Figure 4—figure supplement 2*, *Supplementary file 1j*).

Our research demonstrates that, after only two generations, annual crop plant communities composed of offspring of plants that had been grown in the same community type (as their offspring do now) showed reduced competition and/or increased facilitation compared to plant communities of offspring of plants that had been grown in a different community type (as their offspring do now). In fertilized conditions, common coexistence history also increased overyielding, but this was not the case in unfertilized conditions. Furthermore, common coexistence history had no effect on total yield per plot. We further investigated whether character displacement was responsible for this evolution of plant–plant interactions; contrary to our hypothesis, results did not show evidence for character displacement, but rather for character convergence in plant aboveground traits.

The observed shift in plant–plant interactions towards reduced competition and/or increased facilitation is consistent with a grassland study investigating the effects of community evolution on plant–plant interactions (*Schöb et al., 2018*). However, the lack of response of total yield and biodiversity effects across fertilizing conditions does not agree with several grassland studies examining the effects of common evolution on community productivity and niche differentiation, where it was found that common rapid evolution in plant communities can lead to increases in ecosystem functioning (*van Moorsel et al., 2018*; *Zuppinger-Dingley et al., 2014*; *van Moorsel et al., 2019*; *van Moorsel et al., 2021*; *Meyer et al., 2016*; *Allan et al., 2011*). Only in fertilized plots did we observe a positive effect of common coexistence history on net biodiversity effects (i.e. overyielding), which means that the yield benefit of mixtures compared to monocultures was higher when the plants had been adapted to growing in mixtures (*Figure 3*). Yet we did not observe a significant increase in complementarity effects in response to common coexistence history (*Figure 3—figure supplement 2a*, *Supplementary file 1c*). Surprisingly, selection effects also increased in 4-species mixtures in response to coexistence history (*Figure 3—figure supplement 2b*). This is unexpected, as selection effects have to our knowledge not been shown to increase over time (*Cardinale et al., 2007*). However, it might be that this short common coexistence history has favoured a specific species or a specific trait that was

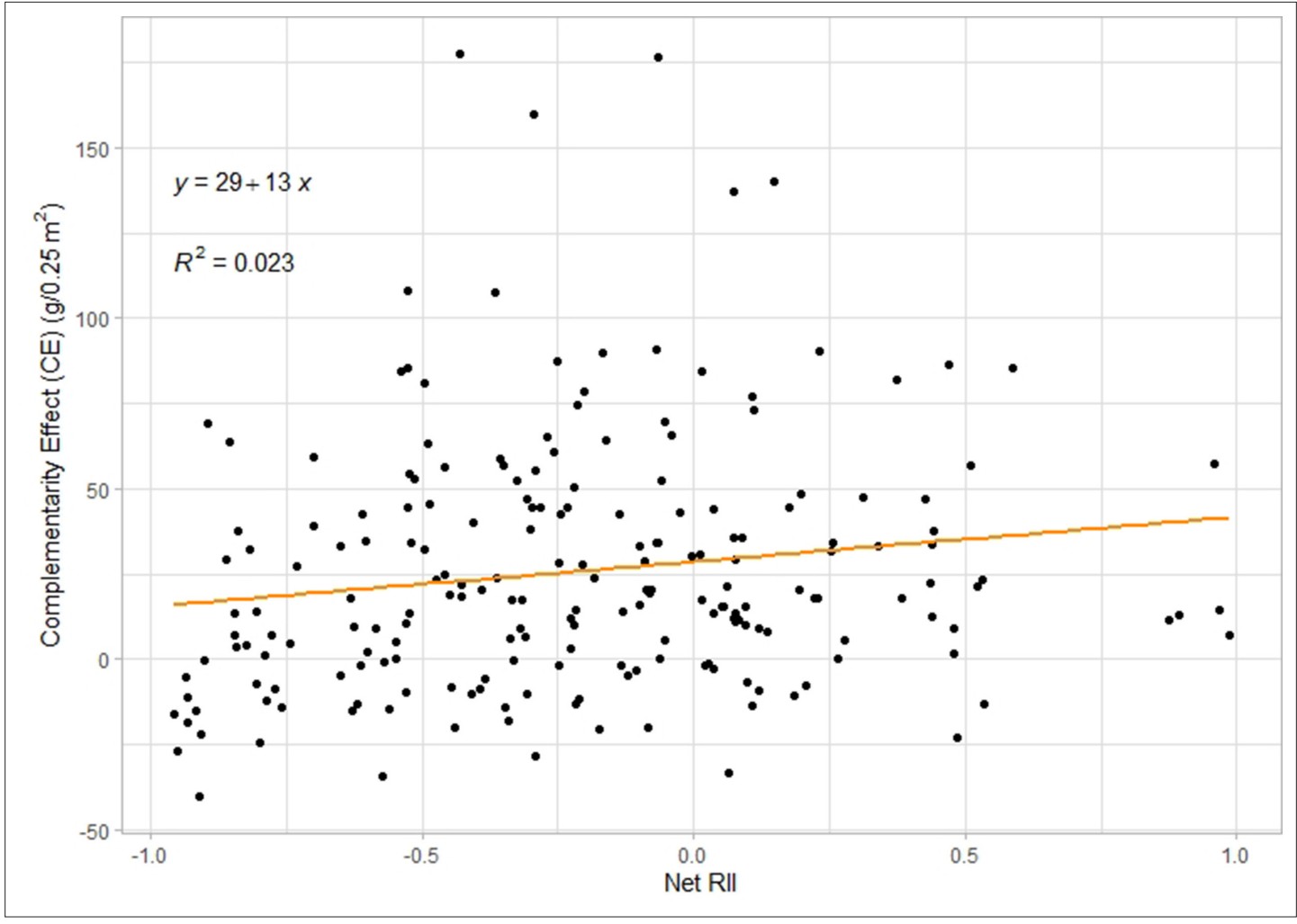

**Figure 5.** Correlation plot between Net RII index and complementarity effect across all plots. There is a significant positive correlation ($F$=4.62, p-value = 0.033, n=204).

The online version of this article includes the following figure supplement(s) for figure 5:

**Figure supplement 1.** Relative effect of coexistence history on single plant growth, with the single plant history as the reference value.

**Figure supplement 2.** Relative effect of coexistence history on monoculture plant growth, with the monoculture plant history as the reference value.

particularly plastic or strongly linked to productivity (*Colom and Baucom, 2021*; *Turcotte and Levine, 2016*).

   The apparent discrepancy between the response of plant–plant interactions and the response of net biodiversity effects to coexistence history can stem from various reasons. First, net biodiversity effects are driven both by complementarity and selection effects *Loreau and Hector, 2001*; therefore, a reduction in competition does not necessarily lead to an increase in net biodiversity effects, as this can be compensated by concurrent changes in selection effects. Changes in RII should however correlate with complementarity effects, which they do in our study (*Figure 5*, p-value = 0.033), indicating that reduced competition and/or increased facilitation correlates with higher complementarity effects. Most importantly though, our RII calculations and net biodiversity effects use different reference levels, that is the single plant vs the monoculture. Indeed, the biodiversity effect calculations ignore the intensity of intra-specific competition and only assess changes in plant–plant interactions from monoculture to mixture, while RII calculations quantify plant–plant interaction intensity in monocultures and mixtures and therefore also allow to assess effects of coexistence history on intra-specific interactions. This can explain why the effect of coexistence history on plant interactions between individuals (quantified through RII) might diverge from the effects of coexistence history on the

diversification of a monospecific community (quantified through the net biodiversity effect, complementarity effect or selection effect). Finally, we also think that the limited timeframe of this study — two generations — might be a reason for the lack of more significant changes in total yield and emphasize the need for longer-term research to confirm the trend identified at the individual level.

Further investigation is also needed to understand the context–dependence of the effects of common coexistence history, notably when reflecting on the important role of fertilization in our results. Our findings are consistent with several recent studies demonstrating that biodiversity effects are higher in high-inputs systems (*Chen et al., 2021*; *Li et al., 2020*; *Stefan et al., 2021b*), and emphasize the role of fertilization in driving yield benefits in diverse crop communities. Indeed, by promoting crop growth and, consequently, higher competition between plants, fertilization may foster higher benefits of niche differentiation — that is reduced competition — in mixtures (*Bertness and Callaway, 1994*; *Stefan et al., 2021a*; *Goldberg and Novoplansky, 1997*).

Overall, increases in biodiversity effects are associated with changes in species traits in response to surrounding plant diversity (*Zuppinger-Dingley et al., 2014*; *Schöb et al., 2018*; *Abakumova et al., 2016*). Traditional hypotheses of trait and niche theory indeed predict that when several species co-occur closely together, selection over generations would favour character displacement that would reduce resource overlap and consequently increase niche differentiation (*Pfennig and Pfennig, 2009*; *Losos, 2000*). Surprisingly, here we found the reverse and observed that a common coexistence history led to a reduction in trait variation, which would suggest a decrease in niche differentiation. Furthermore, functional diversity — calculated as the volume occupied in the space of the traits considered in this study (*Petchey and Gaston, 2002*) — did not respond to common coexistence history (*Figure 6—figure supplement 1*, *Supplementary file 1q*). While surprising, this result is not unheard of *Colom and Baucom, 2021*; *Fox and Vasseur, 2000*; *Grant, 1972*; *Weedon and Finckh, 2021*; notably, because competition for light is asymmetrical, plant height generally converges towards increased plant height in species-rich communities, as a response to a denser and taller canopy (*Lipowsky et al., 1972*; *Falster and Westoby, 2003*). Leaf traits in constrast are usually diverging *Lipowsky et al., 1972*; this was not the case in our study, where we found convergence towards taller plants with lower leaf dry matter content in response to common coexistence history, that is soft leaves associated with rapid biomass production (*Diaz et al., 2016*), and consequently less resource-conservative strategies (*Reich and Cornelissen, 2014*). Lower leaf dry matter content has recently been associated with lower parental or ambient competition (*Puy et al., 2021b*), which is consistent with our results of plant–plant interaction intensities. The traits examined here did not allow to understand the mechanisms behind the observed reduction in competition; we suggest that other traits or processes not measured in this experiment might have responded to the coexistence history treatment. Notably, there could be a shift in belowground traits, such as root-associated traits (*Puy et al., 2021b*), or temporal differentiation of resource capture (*Engbersen et al., 2021*), such as light. We indeed observed a significant increase in light capture ability in communities with a common coexistence history compared to the same communities but with a different coexistence history (*Figure 6*, *Supplementary file 1r*), which indicates that plants used to growing in the same community during several generations might capture the resources more fully than plants coming from a different community. This suggests increased niche differentiation for light use with a common coexistence history. However, here we only rely on our light interception measurements and suggest more longer term studies to understand changes in the use of other resources, such as nutrients or water, and how this is associated to plant traits. Finally, the limited duration of the study as well as the lack of evolutionary potential of some of the chosen crops might also explain why we did not observe clearer signs of increased niche differentiation with common coexistence history.

Furthermore, the scope of this study did not allow us to investigate the transgenerational mechanisms behind these changes in plant–plant interactions and traits in response to coexistence history. The adaptation responses might be genetically based and due to natural selection (*van Moorsel et al., 2019*), as we specifically selected, whenever possible, open-pollinated varieties in order to ensure a maximum amount of genetic variability. This was notably the case for crops that are not standardly used in western European rotations, such as camelina and coriander. Potentially evolutionary mechanisms include sorting out from standing variation, recombination, mutation (*Prentis et al., 2008*), or heritable epigenetic processes (*Rapp and Wendel, 2005*; *Sentis et al., 2018*; *Sobral and Sampedro, 2022*). Because our study only accounted for two generations, recombination and

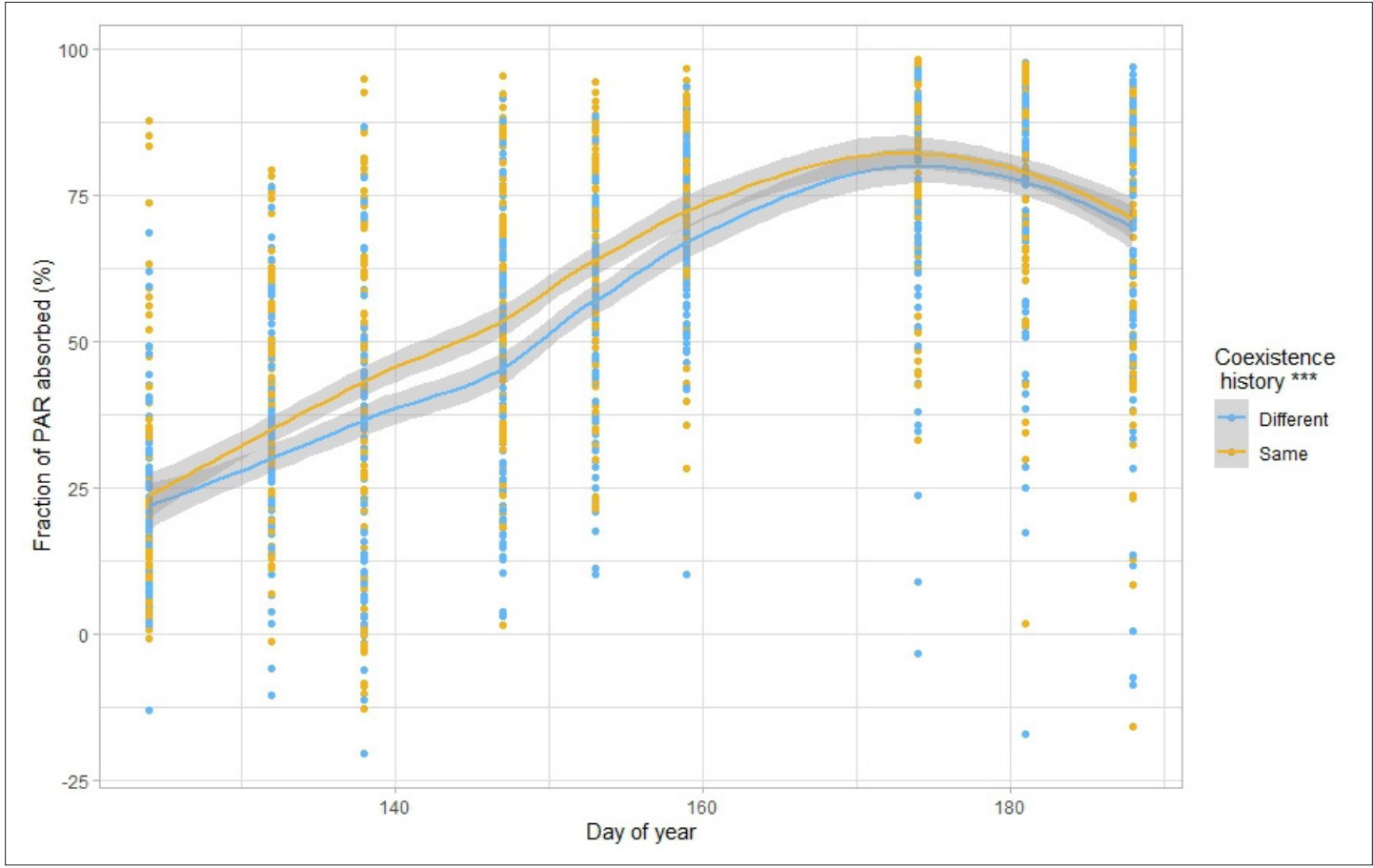

**Figure 6.** Response of absorbed photosynthetically active radiation to coexistence history. n=271. Fraction of PAR absorbed (in %) according to the day of year, for plants with the same or different coexistence history. The lines represent local polynomial regression fittings, with the grey area representing the 0.95 confidence interval. Stars placed next to the legend represent the significance of the result. n=2484. See *Supplementary file 1r* for the complete statistical analysis.

The online version of this article includes the following figure supplement(s) for figure 6:

**Figure supplement 1.** Functional richness in response to crop species diversity, in fertilized and unfertilized plots.

mutation are unlikely, as these are long-term processes (*Hendry et al., 2007*). Rapid adaptation from standing genetic variation is a more plausible mechanism, especially in non-standard species with higher initial variation, such as coriander or camelina (*Prentis et al., 2008*; *Herman and Sultan, 2011*). Particularly, outcrossing could have occurred in the first year of this experiment, as we had a similar experiment running in the same experimental garden with Spanish varieties from the same species (*Chen et al., 2021*; *Stefan et al., 2021a*). However, considering the short timeframe of this study and the low rate of outcrossing in most of our species, we suggest that epigenetic changes — that is stable heritable changes in cytosine methylation — might also have played an important role as potential evolutionary mechanisms (*Verhoeven et al., 2016*; *Sentis et al., 2018*; *Herman and Sultan, 2011*; *Cortijo et al., 2014*; *Van der Graaf et al., 2015*; *Saze et al., 2003*; *Springer, 2013*; *Puy et al., 2021a*). Non-genetic mechanisms can also potentially underpin the observed transgenerational adaptive plasticity; these include seed provisioning, which refers to the carbohydrate, lipid, protein, and mineral nutrient reserves allocated by the maternal plant to the developing seed (*Steets and Ashman, 2010*; *Donohue, 2009*; *Moles and Westoby, 2006*), or changes in maternally derived proteins, mRNAs, or in the relative concentrations of hormones (*Herman and Sultan, 2011*; *Sultan, 1996*; *Dyer et al., 2010*). Finally, a recent study demonstrated the transgenerational role of the seed mycobiome — that is fungal seed–endophytes — for improved resilience and adaptive phenotypes in several generations of wheat *Vujanovic et al., 2019*; thus, heritable transmission of a specific seed mycobiome and/or microbiome might also be a possible non-genetic mechanism (*Vivas et al., 2015*).

For the first time, our study provides empirical evidence for rapid transgenerational adaptation in response to common coexistence history in annual crop communities. Notably, we demonstrated that when plants were growing in the same diversity setting as their parents did for two generations, plant–plant interactions shifted towards reduced competition and/or increased facilitation. This history effect was particularly true for mixtures and was associated with increased overyielding under fertilized conditions. However, there was no significant increase in total yield and no yield benefits in unfertilized conditions. Common coexistence history did surprisingly not lead to character displacement in mixtures, but we instead observed character convergence towards taller plants with lower leaf dry matter content. While further research is needed to assess the validity of our findings in other environmental conditions and for other species, this research emphasizes the importance of considering transgenerational effects of diversity for crop mixtures. This is particularly relevant for breeding programs and highlights the need of including diversity when breeding for crop mixtures, in order to design varieties that could be specifically adapted for intercropping.

## Methods

### Study sites

The Crop Diversity Experiment took place in 2018, 2019, and 2020 in an outdoor experimental garden located at the Irchel campus of the University of Zurich, Switzerland (47.3961 N, 8.5510 E, 508 m a.s.l). Zurich is characterized by a temperate climate (*Stefan et al., 2021a*). The experimental garden was irrigated during the growing season with the aim of maintaining a sufficient amount of water for optimal plant growth. The dry threshold of soil moisture was set at 50% of field capacity, with a target soil moisture of 90% of field capacity. Whenever dry thresholds were reached measured through PlantCare soil moisture sensors (PlantCare Ltd., Switzerland), irrigation was initiated, and water added until reaching the target value.

Each experimental garden consisted of square plots of 0.25 m$^2$. The uppermost 30 cm of the square plots were filled with standard, not enriched, agricultural soil coming from the local region. This soil consisted of 45% sand, 45% silt, and 10% clay, and initially contained 0.19% nitrogen (N), 3.39% carbon (C), and 332 mg total phosphorous (P)/kg, with a mean pH of 7.25. Beneath that, there was local soil of uncharacterized properties that allowed unlimited root growth. The plots were embedded into larger beds of 7x1 m, each bed containing 28 plots. Inside a bed, plots were separated from each other by metal frames. The metal frames reached 10 cm aboveground and until 30 cm belowground. While the relatively small plot sizes allowed us to undertake a large experiment under environmentally highly controlled but realistic outdoor conditions, some variables can suffer edge effects and interferences with neighbouring plots. However, such effects would probably increase residual variation more than between-treatment variation, because randomization was used to prevent confounding of between-plot interactions with treatments. In the only relevant study of which we are aware, the biodiversity–productivity relationship in herbaceous communities was not affected by plot size (*Roscher et al., 2005*) while a recent theoretical study showed that, if anything, biodiversity effects should increase with plot size (*Isbell et al., 2018*).

We therefore assume that effect size in our experiment, if anything, is probably rather conservatively estimated compared with that in studies using larger plot sizes.

Every year, we fertilized half of the beds with N, P and potassium (K) at the concentration of 120 kg/ha N, 205 kg/ha P, and 120 kg/ha K. Fertilizers were applied three times per year, namely once just before sowing (50 kg/ha N, 85 kg/ha P, 50 kg/ha K), once when wheat was at the tillering stage (50 kg/ha N, 85 kg/ha P, 50 kg/ha K), and once when wheat was flowering (20 kg/ha N, 34 kg/ha P, 20 kg/ha K). The other half of the beds was left unfertilized. In 2018, we randomly allocated individual beds to a fertilized or non-fertilized treatment. In the following years, we kept the initial fertilization treatment allocation.

### Crop species

Experimental communities were constructed with six annual crop species of agricultural interest. We selected only seed crops with similar growth requirements in terms of climate and length of growing season, and with similar plant sizes to fit at least 40 individuals in the rather small plots. The six species belong to four different phylogenetic groups with varying functional characteristics: we first separated

**Table 1.** List of crop species ecotypes and their suppliers.
*Avena sativa* (oat) is mainly self-pollinating, with outcrossing rates of around 1% (*Shorter et al., 1978*). The variety Canyon was acquired in 2014 through conventional selection processes.

| Species | Switzerland | |
| --- | --- | --- |
| | Ecotype | Supplier |
| *Avena sativa* | Canyon | Sativa Rheinau |
| *Triticum aestivum* | Fiorina | DSP, Delley |
| *Coriandrum sativum* | Indian | Zollinger Samen, Les Evouettes |
| *Lens culinaris* | Anicia | Agroscope, Reckenholz |
| *Camelina sativa* | n.a. | Zollinger Samen, Les Evouettes |
| *Linum usitatissimum* | Lirina | Sativa Rheinau |

monocots [*Triticum aestivum* (wheat, C3 grass, Poaceae) and *Avena sativa* (oat, C3 grass, Poaceae)] and dicots. Among the dicots, we differentiated between suparasterids [*Coriandrum sativum* (coriander, herb, Apiaceae)] and superrosids. Among the superrosids, we separated legumes [*Lens culinaris* (lentil, legume, Fabaceae)] from non-legumes [*Linum usitatissimum* (flax, herb, Linaceae) and *Camelina sativa* (false flax, herb, Brassicaceae)]. Furthermore, we chose crop varieties that were locally adapted and commercially available in Switzerland (*Table 1*).

*Triticum aestivum* (wheat) is principally self-pollinating, with outcrossing rates generally between 1 and 4% (*Hai et al., 2005*; *Loureiro et al., 2012*), although some cultivars have been shown to have outcrossing rates up to 8% (*Lawrie et al., 2006*). Fiorina is an accession originating from Switzerland, acquired in 2015, specifically for organic agriculture.

*Coriandrum sativum* (coriander) has a generally high genetic variability, with studies showing up to 70.46% polymorphism, indicating the presence of high degree of molecular variation in the studied coriander varieties (*Choudhary et al., 2019*; *Singh et al., 2012*). The variety that we used originally came from an Indian market and was not a fixed variety, which ensured a minimum of genetic variability. The flowers of coriander are self-incompatible but plants are self-compatible. Geitonogamy is therefore common. Cross-pollination is facultative but can reach up to 20% (*Diederichsen, 1996*).

*Lens culinaris* (lentil) is mainly self-pollinating; depending on the cultivar, outcrossing rates reach between 1 and 5% (*Horneburg and Weber, 2006*).

*Camelina sativa* (camelina) is mainly self-pollinating, with outcrossing rates of less than 1% (*Walsh et al., 2006*; *Walsh et al., 2015*). In the study, we used a local landrace that was not a fixed variety.

*Linum usitatissimum* (flax) is mainly self-pollinating but outcrossing does occur, at a rate of 1–5% (*Jhala et al., 2011*). Lirina, the variety of Linum that we used has been defined by ProSpecieRara as a rare or ancient variety. ProSpecieRara ensures the preservation of rare traditional varieties (*Begemann, 2002*). Furthermore, studies have shown that linseed varieties have higher genetic variability than fiber flax and should therefore be considered as valuable genetic resources (*Vromans et al., 2006*; *Hoque et al., 2020*).

## Experimental crop communities

Experimental communities consisted of single plots with one individual, monocultures, 2- and 4-species mixtures (*Figure 1*, *Figure 1—figure supplement 1*). We planted every possible combination of 2-species mixtures with two species from different phylogenetic groups and every possible 4-species mixture with a species from each of the four different phylogenetic groups present (*Supplementary file 1*). We replicated the experiment two times with the exact same species composition, except for single individuals which were replicated 4 times. Single plants were allocated to separate beds in order to minimize interference among neighbouring plots (*Figure 1*), and randomized within each fertilized treatment. Monoculture and mixture plots were randomized among plots and beds within each fertilizer treatment. Each monoculture and mixture community consisted of one, two or four species planted in four rows. Two species mixtures were organized following a speciesA|speciesB|speciesA|speciesB pattern. The order of the species was chosen randomly. Four species mixtures were organized following a speciesA|speciesB|speciesC|speciesD pattern. The order of the species

was also randomized for each 4-species mixtures to avoid having the same order of species for all the replicates of a same mixture. Density of sowing differed among species groups and was based on current cultivation practices: 160 seeds/m$^2$ for legumes, 240 seeds/m$^2$ for superasterids, 400 seeds/m$^2$ for cereals, and 592 seeds/m$^2$ for superrosids. These correspond to the densities in monocultures; in mixtures, we kept these densities for each species (e.g. for legume, we planted 10 individuals per line in the monocultures and also 10 individuals per line in the mixtures). Each year, seeds were sown by hand in early April.

## Adaptation treatment

In 2019, we used the seeds collected in 2018 to add a coexistence history treatment: we repeated the experiment with seeds coming from single individuals, monocultures, and mixtures, respectively. This means that each plot described above was repeated three times: once with seeds coming from single plants, once with seeds coming from monoculture plants, and once with seeds coming from mixture plants. We respected the fertilizing treatment, that is there was a history treatment for each fertilizing condition. When planting the mixtures with a mixture history, we specifically used seeds coming from the same species combination. When planting the monocultures and singles with a mixture history, we used seeds coming from a common pool combining all 4-species mixtures. Plots were fully randomly re-allocated each year to avoid soil legacy effects.

In 2020, we repeated this process and selected seeds from 2019 to sow the single and community plots. We only selected seeds that had a 'pure' history, that is that were always grown in the same coexistence history (for instance, for single history seeds in 2020 we selected only seeds that were grown as singles also in 2018 and 2019).

## Data collection

### Photosynthetically active radiation (PAR)

Interception of PAR by the plant canopy was measured weekly with a LI-1500 (LI-COR Biosciences GmbH, Germany). In each plot, three PAR measurements were taken around noon by placing the sensor on the soil surface in the center of each of the three in-between rows. Light measurements beneath the canopy were compared to ambient radiation through simultaneous PAR measurements of a calibration sensor, which was mounted on a vertical post at 2 m above ground in the middle of the experimental garden. FPAR (%) indicates the percentage of PAR that was intercepted by the crop canopy.

### Traits measurements

At the time of flowering, three individuals per crop species per plot were randomly marked. We measured the height of each individual with a ruler from the soil surface to the highest photosynthetically active tissue. We then measured plant width with a ruler by taking the largest horizontal distance between two photosynthetically active tissues. We sampled one healthy leaf from each marked individual and immediately wrapped this leaf in moist cotton; this was stored overnight at room temperature in open plastic bags. The following day, we removed any excess surface water on the leaf and weighed it to obtain its water saturated weight (*Cornelissen et al., 2003*). Then this leaf was scanned with a flatbed scanner (CanoScan LiDE 120, Canon), oven-dried in a paper envelope at 80 °C for 72 hr, and subsequently reweighed to obtain its dry weight. We calculated Leaf Dry Matter Content (LDMC) as the ratio of leaf dry mass (g) to water saturated leaf mass (g). Using the leaf scans, we measured leaf area with the image processing software ImageJ (*Schneider et al., 2012*). Specific Leaf Area (SLA) was then calculated as the ratio of leaf area (cm2) to dry mass (g).

### Plot grain yield and biomass

Grain yield and aboveground biomass of each crop species was determined per plot at maturity. This corresponded to July/August. As time of maturity slightly varied among the different crop species, we conducted harvest species by species. We clipped plants right above the soil surface and separated seeds from the vegetative parts. Seeds were sun-dried for 5 days and weighed. Biomass was oven-dried at 80 °C until constant weight and weighed.

## Individual yield and biomass

We harvested the three marked individuals for the trait measurements separately; we separated seeds from aboveground biomass and they were both dried and weighed as previously mentioned. Furthermore, for each marked individual we weighed ten randomly selected seeds to obtain the mass per seed.

## Data analyses

All analyses were performed using R version 4.1.0 (*R Development Core Team, 2019*).

### Plant Interaction Index

Plant interaction intensity in the plots was calculated for each marked individual by means of the relative intensity index (RII) defined as such (*Diaz Sierra et al., 2017*):

$$RII = \frac{yield_{comm} - yield_{single}}{yield_{comm} + yield_{single}} \tag{1}$$

, where $yield_{single}$ is the grain yield (in grams) of a single plant grown in isolation, and $yield_{comm}$ is the grain yield (in grams) of an individual of the same species when grown in a community. $yield_{single}$ was calculated for each species, fertilizing conditions and coexistence history by taking the average of the four corresponding replicates. RII is a standardized index with commutative symmetry commonly used to measure plant–plant interactions (*Armas et al., 2004*). A positive RII means that the individual is benefiting — in terms of productivity, that is yield — from being in a community compared to growing alone, and therefore indicates facilitation. On the contrary, a negative RII means that the individual is suffering from being in a community compared to growing alone, and therefore indicates competition. RII values of all species (*a,b,c,d*) composing the community (i.e. species a in case of a monoculture and species a to d in case of a mixture of four species) were averaged and subsequently weighted by their relative abundance $r_i = \frac{1}{number\ of\ species}$ to calculate the mean net interaction in the community (RIInet):

$$RIInet = \sum_{i=a}^{d} \left( RII_i r_i \right) \tag{2}$$

This net index thus indicates whether on the community level, plants are experiencing facilitation or competition. The closer this index gets to 1, respectively –1, the stronger the facilitation, respectively competition. To check the applicability of this net index, we looked at the correlation between this index and the complementarity effect from Hector & Loreau (*Vandermeer, 1992*) (see below for the calculations) and indeed we found a positive correlation across all plots (*Figure 5*, [*F*=4.62, p-value = 0.033, n=204]). This shows that a higher net index — that is decreased competition — indeed correlates with higher complementarity effects and therefore, we are confident that net RII reasonably describes plant interactions.

The reference values for RII per species were computed per fertilizer and coexistence history, which means that each coexistence history has a different reference value. We chose this way of calculating these metrics as this allows to explicitly distinguish the effects of coexistence history on the interactions, independently of the baseline effect on plant performance. This follows the classic framework of plant–plant interaction and facilitation work (*Michalet et al., 2014*). To further investigate potential changes in reference plant performance, we calculated *RII coexistence* for each community (i.e. for single plants, for monocultures, and for mixtures) using the following calculations.

$$RII\ coexistence = \frac{yield_{single\ with\ community\ history} - yield_{single\ with\ single\ history}}{yield_{single\ with\ community\ history} + yield_{single\ with\ single\ history}}$$

for single plants and

$$RII\ coexistence = \frac{yield_{monoculture\ with\ mix\ or\ single\ history} - yield_{monoculture\ with\ mono\ history}}{yield_{monoculture\ with\ mix\ or\ single\ history} + yield_{monoculture\ with\ mono\ history}}$$

for monocultures (*Figure 5—figure supplements 1 and 2*).

## Net biodiversity effect

For all mixture communities we quantified the net biodiversity effect (NE) defined as the overyielding relative to the expected yield based on monocrop values.

$$\text{NE} = \Delta Y = Y_o - Y_E = Y_o - \sum_{i=1}^{s} \left( r_i M_i \right)$$

where $Y_o$ is the observed yield of the mixture, $Y_E$ is the expected yield measured as the sum of the monocrop yield of each species ($M_i$ weighted by the species proportion in the mixture. The monocrop yield was calculated for each species, fertilizing conditions and coexistence history by taking the average of the two corresponding replicates.

We partitioned net biodiversity effect into its two components, the complementarity and selection effects according to *Loreau and Hector, 2001*.

$$\text{NE} = N \cdot \overline{\Delta RY} \cdot \overline{M} + N \cdot cov \left( \Delta RY, M \right) \qquad (5)$$

where N is the number of species in the plot, ΔRY is the deviation from expected relative yield of the species in mixture in the respective plot, which is calculated as the ratio of observed relative yield of the species in mixture to the yield of the species in monoculture, and M is the yield of the species in monoculture. The first component of the biodiversity effect equation ($N \cdot \overline{\Delta RY} \cdot \overline{M}$) is the complementarity effect (CE) and represents how much individual species contribute more to productivity than predicted from monoculture. The second component ($N \cdot cov \left( \Delta RY, M \right)$) is the selection effect (SE) and describes the greater probability of more diverse communities including highly productive species which account for the majority of productivity.

## Total crop yield

To assess crop performance, we calculated total crop yield per plot as the sum of total seed mass per species.

## Trait analyses

Traits were analysed both at the species-level and at the plot-level. At the species level, we calculated the mean and coefficient of variation (CV) per species for each trait per plot. At the plot-level, we calculated Community-Weighted-Means (CMW, which is defined as the average of trait values for each species weighted by the species relative biomass *Miller et al., 2004*), and coefficient of variation per plot for each trait.

Functional richness (FRic) was calculated in each plot using the function *dbFD* from the package *FD* (*Laliberté and Legendre, 2010*), by measuring the convex hull volume occupied by the individuals of a plot in the space of the considered traits.

To analyze the effects of the experimental treatments on RIInet, NE, CE, SE, total crop yield, FRic, and CWM and CV per plot, we used generalized linear mixed models using the function *lmer*. Fixed factors included fertilizing condition (yes or no), coexistence history (considered as 'same' or 'different'), crop species number (2 vs 4) nested in monoculture vs mixture, as well as the interactions between them. Species composition, bed and column were set as random factors.

*Response variables* (*e.g. yield*) *fertilization* ∗ *coexistence history*∗

(*mono vs mix + crop species number*) + (1|*comb*) + (1|*bed*) + (1|*column*)

Effect sizes were calculated from marginal means obtained using the function *emmeans,* and pairwise comparisons were calculated using Tukey tests from the *emmeans* function (*Lenth, 2021*). To analyze the effects of the experimental treatments on the mean and coefficient of variation of the different traits per species (height, width, SLA, LDMC, mass per seed, respectively), we used generalized linear mixed models using *lmer* with the same fixed factors as previously described. Species, species composition, bed and columns were set as random factors. The response variables were log-transformed or square-root-transformed where needed. To analyse the response of FPAR, we calculated the average of the three measurements per plot for each week and analysed its response by using similar linear mixed models as described above on all the dates, with day of year as a random

factor. For all models, we tested for normality of the residuals using a Shapiro–Wilk test and homogeneity of the variance using a Levene test.

## Acknowledgements

We thank Elisa Pizarro Carbonell, Carlos Barriga Cabanillas, Anja Schmutz, Sandra Gonzalez Sanchez, Lukas Meile, Carlos Federico Ingala, Roman Hüppi, Simon Baumgartner, Benjamin Wilde, Manon Longepierre, Marijn Van de Broek, Leonhard Späth, Inea Lehner, Anna Bugmann, Jianguo Chen, Nicola Haggenmacher and Zita Sartori for their help in the field, and Johan Six for comments on the experimental design. We also thank the Aprisco de Las Corchuelas Field Station and the University of Zurich for the use of their facilities. The study was funded by the Swiss National Science Foundation (PP00P3_170645).

## Additional information

### Funding

| Funder | Grant reference number | Author |
| --- | --- | --- |
| Swiss National Science Foundation | PP00P3_170645 | Christian Schöb |

The funders had no role in study design, data collection and interpretation, or the decision to submit the work for publication.

### Author contributions

Laura Stefan, Conceptualization, Data curation, Formal analysis, Investigation, Visualization, Writing - original draft, Writing - review and editing; Nadine Engbersen, Data curation, Validation, Investigation; Christian Schöb, Conceptualization, Resources, Data curation, Formal analysis, Supervision, Funding acquisition, Validation, Investigation, Methodology, Project administration

### Author ORCIDs

Laura Stefan  http://orcid.org/0000-0003-0798-9782

### Decision letter and Author response

Decision letter https://doi.org/10.7554/eLife.77577.sa1
Author response https://doi.org/10.7554/eLife.77577.sa2

## Additional files

### Supplementary files

• Transparent reporting form

• Supplementary file 1. Supplementary statistical analyses. (a) Type-I Analysis of Variance Table of the experimental treatment effects on net, competition and facilitation indexes (RII), in year 3 (2020). (b) Pairwise comparisons of the effect on net interaction index (RII) between fertilizer (yes, no), coexistence history (diff [different], same), and monoculture vs mixture (mix [mixture], mono [monoculture]). (c) Type-I Analysis of Variance Table of the experimental treatment effects on net, complementarity, and selection effects in year 3 (2020). (d) Pairwise comparisons of the effect on net biodiversity effects between fertilizer (yes, no) and coexistence history (diff [different], same). (e) Pairwise comparisons of the effect on selection effects between fertilizer (yes, no), coexistence history (diff [different], same), and planted diversity (2 vs 4). (f) Type-I Analysis of Variance Table of the experimental treatment effects on total crop yield per plot (square-root transformed). (g) Type-I Analysis of Variance Table of the experimental treatment effects on mean and coefficient of variation of height, per species per plot (species level) in year 3 (2020). (h) Type-I Analysis of Variance Table of the experimental treatment effects on mean and coefficient of variation of width, per species per plot (species level) in year 3 (2020). (i) Type-I Analysis of Variance Table of the experimental treatment effects on mean and coefficient of variation of SLA, per species per plot (species level) in year 3 (2020). (j) Type-I Analysis of Variance Table of the experimental treatment effects on

mean and coefficient of variation of LDMC, per species per plot (species level) in year 3 (2020). (k) Type-I Analysis of Variance Table of the experimental treatment effects on mean and coefficient of variation of mass per seed, per species per plot (species level) in year 3 (2020). (l) Type-I Analysis of Variance Table of the experimental treatment effects on community-weighted mean and coefficient of variation of height, per plot (community level) in year 3 (2020). (m) Type-I Analysis of Variance Table of the experimental treatment effects on community-weighted mean and coefficient of variation of width, per plot (community level) in year 3 (2020). (n) Type-I Analysis of Variance Table of the experimental treatment effects on community-weighted mean and coefficient of variation of SLA, per plot (community level) in year 3 (2020). (o) Type-I Analysis of Variance Table of the experimental treatment effects on community-weighted mean and coefficient of variation of LDMC, per plot (community level) in year 3 (2020). (p) Type-I Analysis of Variance Table of the experimental treatment effects on community-weighted mean and coefficient of variation of mass per seed, per plot (community level) in year 3 (2020). (q) Type-I Analysis of Variance Table of the experimental treatment effects on functional richness in year 3 (2020). (r) Type-I Analysis of Variance Table of the experimental treatment effects on FPAR in year 3 (2020). (s) List of species mixture combinations.

## Data availability

The data that support the findings of this study are available on Zenodo: https://doi.org/10.5281/zenodo.5223410.

The following dataset was generated:

| Author(s) | Year | Dataset title | Dataset URL | Database and Identifier |
|---|---|---|---|---|
| Stefan L | 2021 | Rapid adaptation in Intercropped Systems | https://doi.org/10.5281/zenodo.5223410 | Zenodo, 10.5281/zenodo.5223410 |

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
