## [Editor Report]

This study reports that interactions between crop species grown over three generations in mixture instead of monoculture become less competitive/more facilitative, suggesting a way to breed for increased mixture yield. This fundamental finding is of high interest to the fields of ecology and agriculture, as well as society seeking new solutions to satisfy increasing food demands. The methodological approach is compelling, yet distinguishing between reduced competition and increased facilitation remains challenging.

---

## [Decision Letter]

**Decision letter after peer review:**

Thank you for submitting your article "Rapid transgenerational adaptation in response to intercropping reduces competition" for consideration by *eLife*. Your article has been reviewed by 4 peer reviewers, and the evaluation has been overseen by a Reviewing Editor and Meredith Schuman as the Senior Editor. The following individual involved in the review of your submission has agreed to reveal their identity: Piter Bijma (Reviewer #3).

The reviewers agree that the idea behind the paper is a good one and the experiment is appropriate and well analyzed.

Essential revisions:

In your revision, please respond point-by-point to these essential revisions, which all reviewers agreed upon. In addition, the individual reviews, which are provided below, may contain some further points of interest.

1) The major result as suggested in the title seems to be reduced competition of offspring with a co-occurrence history. However, reviewers feel that results about biodiversity and yield effects should be the focus: greater weight is given to results on RII, whereas results on NBE/yield are only marginally considered. The emphasis should rather be put on NBE and yield. They are the variables of interest to characterize the consequences of plant-plant interaction at the group level and they are the ones classically used in the BEF field (see Zuppinger-Dingley et al., Nature, 2014).

2) However, the significance of the effect of coexistence history on NBE in the fertilized treatment is not apparent in the manuscript. The analysis of fertilized vs. unfertilized treatments should be improved to better show if indeed there is an indication of greater net biodiversity effects under fertilized conditions. The analysis of traits should also be improved by separating between- from within-species variation and referring to work that shows different traits respond differently (height generally converging and leaf traits generally diverging, work by Roscher et al.).

3) The discrepancy between 1) and 2) is very hard to understand and must be discussed by the authors. The way the single-plant reference is used for RII may affect the interpretation: is the apparent reduction in competition due to the mixture coexistence history, or due to the single reference plant? Because competition is always measured in terms of yield effects, the RII and total yield results would seem to rely on the same data. Hence, how can there be a true decrease in competition without an increase in yield? If this does not become clear, then the paper lacks a clear message.

It is quite difficult to interpret RII and NBE as we do not know how the reference values were computed and if the same reference values were used across treatments. Because these indices are relative to single plants or monocultures, changes in index values can be caused by changes in the reference values (i.e. single plant or monoculture, respectively), not only by changes in the value measured in the mixtures.

4) Compared with the SGH, there is as much evidence for the opposite response of increased biodiversity effects with improved environmental conditions. The authors should discuss their absence of evidence for SGH.

5) The authors should tone done facilitation and focus on reduced competition (as they do in the title and abstract). They should probably omit the partitioning of the competition index into competition and facilitation components. For example, when RII goes from -0.5 to -0.3 using the same single isolated plant yield as a reference, it means that single plant yield has increased in the mixtures. This can be caused either by higher facilitation or lower competition. How can you tell the two mechanisms apart?

6) Results for the second year might be included in the SI.

7) Potential mechanisms underpinning transgenerational effects and reduced competition should be better explained. Thus, for the first it should be mentioned that it could occur by evolution via sorting out from standing variation (for highly selfing species if at least there was variation between initially sown genotypes), recombination (probably minor contribution in only 2 generations) or mutation (even less likely) or by epigenetic/physiological carry-over processes.

8) The methods should be described in more detail.

*Reviewer #1 (Recommendations for the authors):*

I think the manuscript can be improved by removing the first index (RII) and focusing on plot-level analysis. This RII is not commonly used and not very explicit for the reader compared to NBE (or NE). Overall, the plot level data and trait data point to the same direction, which is no increase in complementarity between species over a generation of coexistence. I would focus on this message and discuss the limits of the design to detect a potential increase in complementarity if they were to occur as described in grasslands (inbred lines with low evolutionary potential, short evolutionary time, alternate rows which limit interactions, etc).

I would avoid commenting on "trends" and non-significant results in the "Results and Discussion" section. I would also add some significance measures in this section (either p-value or stars on the Figures, or ANOVA Tables).

Line by line comment:

l. 15: Why using seeds selected in monoculture could compromise yield benefit in mixtures? This is explained in the following sentence, but we lack the connection with this sentence.

l. 37: ref Vandermeer, J.H. (1992) The Ecology of Intercropping, Cambridge University Press.

l. 44-47: ref Meilhac, J. et al. (2020) Both selection and plasticity drive niche differentiation in experimental grasslands. Nat. Plants 6, 28-33.

l. 44: The first sentence is very general and does not convey any information. In the second sentence, it needs to be explained why the use of commercial seeds bred for monocultures might not be optimal to promote positive diversity effects.

l. 52: "changed and evolved" is redundant.

l. 54: I would provide more information on the species (species or functional groups) and say if they are commonly grown as intercrops.

l. 56: mesocosms is only used here and not defined.

l. 57-8: "We selected open-pollinated varieties", This does not seem to be the case for oat, wheat, and lentil in the Methods section.

l. 62: It is not clear what is the difference between this Relative Interaction Index, and the classical Relative Yield Total used in the partitioning of Loreau and Hector at this stage. Why use both? What is the information added by Relative Interaction Index?

l. 69 to 79: it should be stated that this hypothesis applies to mixture communities only.

l. 77-8: The SGH arrives a bit "out of the blue" here. It needs to be defined and developed within the context of intercropping before.

l. 81-82: Figure 2 and Table S1 suggest that the "history" treatment effect is primarily driven by facilitation, not by competition. Put differently, the difference between "same" and "different" evolutionary history is much bigger for the positive interaction index than for the negative ones. The sentences here suggest the opposite, i.e. greater effect on competition than on facilitation.

l. 83 and throughout the manuscript: "SI" can be removed before supplementary material references.

l. 91: "less competition": I think it should be "more facilitation" instead (cf previous comment).

l.99 to 105: this whole part only comments on non-significant results.

l. 112: I could not find to which data and test the p-value and F value reported here refer.

l. 142-144: "height community-weighted mean was not significantly different between coexistence histories".

Paragraph 120-145: I would add the results on Functional hypervolume and PAR here. I think it is especially interesting to discuss the differences in PAR interception between communities with different coexistence histories (you do have significant evidence of complementary effects increased in mixtures with a mixture history here).

l. 172-173: "In fertilized conditions, this shift in plant-plant interactions was associated with an increase in overyielding": I could not find the statistics supporting that statement. In Table S4, the Fertilizer x history effect is significant, but we do not have the tests of the history effect within each fertilization treatment. Also, the p-value used to support this in the result section (l. 103) is higher than 0.05.

l. 194-200: this discussion should be confronted with the Stress-Gradient Hypothesis, as presented in the Introduction. Indeed, the SGH predicts the opposite pattern as the one observed in the study, i.e., stronger biodiversity effects in low-input systems.

l. 205-206: "a reduction in trait variation favoured increased yield benefits in mixtures": I cannot find any result supporting that statement.

l. 209: "plants might have adapted to express the phenotype that would maximise their fitness": again, I do not see any relationship between traits and fitness in the results.

l. 218-222: this result should be presented in the main text.

l. 227-228: "we specifically selected open-pollinated varieties in order to ensure a minimum amount of genetic variability". There are two problems here. First, it was not open-pollinated varieties for all species. Second open-pollinated varieties have more genetic variability than other varietal types such as inbred lines or hybrids.

l. 262: 30 cm of what?

l. 268: were the metal frames only belowground or also aboveground?

l. 286 and the whole sub-section: It is not easy to assess the extent of genetic variation within each species in this section. It seems that there was no within-species variation for wheat, oat, and lentil, and maybe some variation for the 3 others. I think having this kind of information would be important to interpret the results phenotypic changes can occur both through selection and plasticity, depending on the amount of standing variation in the species.

l. 335 to 337: I first did not understand that single plant plots were grouped together in separate beds. In fact, I only got it later with the picture in Figure S10. I would reword this part.

l. 340-342: We have no information on how the different species were arranged in 4-species mixtures.

l. 342-343: Are these densities the monoculture densities? If yes, how were these densities adjusted in mixtures?

l. 400: "yield": is it grain or biomass yield? Also, how was "Yield single" computed? Did you consider only a single plant from the same year, same fertilization, with a pure single plant history? Were the different single plants from the same species averaged, corrected for design effects, etc?

l. 410: "proportional": amend to "relative".

l. 418-420: Why not use directly the Loreau and Hector partitioning then?

l. 431 to 436: Since RIINet and NInC are highly correlated, it is not surprising that you get the same results: you just repeated your analysis with a redundant index. I would either remove or just mentioned that RIInet and NIntC were strongly correlated and thus provided identical results.

l. 440: as for RII and single plant yield, how were monoculture yields computed here? Was it averaged across replicates of the same species, only considering replicates with a "pure" monoculture history?

l. 451-452: Community-Weighted Mean needs to be defined and referenced.

l. 456 and 457: "NIntC" should be amended to "RII" and "LER" should be amended to "SE"

Table S17: Which PAR values were used in this analysis of variance? Was it a single date, or averaged across dates?

Figure S10: I would move this Figure to the main text or provide an illustration of the experimental set-up there to help the readers visualize what are beds, plots, and how are the different species arranged, notably in mixture plots.

*Reviewer #2 (Recommendations for the authors):*

1. Please give a more balanced discussion of the proposed mechanisms for how co-occurrence can lead to more facilitation. I see how co-occurrence could lead to trait displacement and less niche overlap, so less competition. But what is the facilitation part of this? Please include something in the intro that covers this.

2. Your RII_fac and RII_comp appear to be superficially separated based on whether the net effects of neighbors were positive or negative, is this correct? What would happen if one species in a mixture plot had net positive effects of neighbors and another species had net negative effects of neighbors? Would you calculate the whole plot as just the sum of that? Or how is it calculated in that case?

3. Also, how do you separate intraspecific competition and interspecific competition from these metrics? For example, if intraspecific comp > interspecific comp, you would still see a less negative RII_comp at higher diversity. But this isn't necessarily because competition is actually weaker at higher diversity (but instead a shift from one type of competition to another). Similarly, if intraspecific comp > interspecific comp, you could also see these shifts in RII_fac. A stronger RII_fac value could be completely driven by just alleviation of intraspecific competition at higher diversity.

4. I'm thus consistently tripped up by the many phrases in the MS where you say things like: "shifted towards weaker competition, and in some cases, stronger facilitation" (lines 81-82). All of your RII values are net effects, so how can you conclude that what drives the changes you see is due to competition or facilitation? Why not: "shifted towards weaker competition and/or stronger facilitation (though teasing out the differences is not possible in our given dataset)".

5. Overall trends I see in RII: most things do worse when growing next to same species neighbors (vs. alone), but the same or better if they grow next to heterospecific neighbors. This gets stronger if they have coexisted for multiple generations. And the positive aspects of it are most strongly affected by coexistence. Your data indicate that this ISN'T due to trait divergence which is super interesting! You also show a strong correlation with selection effects which indicates that a couple of species are driving the patterns. Did you take a look at how selection effects were correlated with RII? Do you know which species are doing that? Have you split into plots with and without legumes? Do legume effects on neighbors get stronger with coexistence history? The only other mechanism I could imagine is more specialization of enemies over time leading to stronger dilution effects? Please discuss these possible mechanisms in some way.

6. The loss of variability in each species when they have coexisted for multiple generations also seems to point towards some kind of restricted gene flow. How much interbreeding was there between plots? Is there something about the environment that you could imagine selecting for these very tall plants? Also, the fact that they are getting more like each other overall, but still having more positive effects on each other is very strange. Need more context about how this might be explainable.

*Reviewer #3 (Recommendations for the authors):*

Background on main comments / further general comments

The presence of clear effects of the coexistence history on competition together with absence of effects on yield is worrying. Particularly because effects on competition relative to single plants (Figure 2) are statistically very significant while those on Net Effect relative to monocrop are borderline (P = 0.0715). At the same time, the plot of total yield excludes the results for single plants (these should be included).

The decrease in competition due to the same coexistence history shown in Figure 2 is relative to single plants. Hence, the comparison is relative. Do we really see a decrease in competition here, or is the single plant getting worse due to the coexistence history? Importantly: It is unclear in Figure 2a whether the two points that make up a pair of points for the same x-axis value are relative to a single plant with the same coexistence history. For example, for x = 1 (monoculture) the two y-axis values are about -0.3 and -0.2. Is the reference single plant the same for these two points the same, so that we can also infer that the difference between both points is 0.1? Or is the reference single plant for the first point (-0.3) a single plant with also a Different co-existence history, and the reference single plant for the second point (-0.2) a single plant with also a Same co-existence history. If the latter is the case, then both points have a different reference, and the apparent difference of 0.1 may very well originate from having two different single plant references, rather than an effect on competition in the monocrop.

Related to the previous comment: I cannot reconcile the results for No Fertilizer in Figures 3a and 3b. NE is a measure of overyielding relative to monocrops. Figure 3a NoFertilizer shows that overyielding is greater with Different. However, 3b shows greater total yield with Same. This strongly suggests that the two points making up a pair in Figure 3a do not have the same monocrop as the reference. Hence, then we don't know whether the effect is due to the coexistence history of the 2-mixture or due to the co-existence history of the monocrop used as reference. Things get very confusing because the reference point seems to be shifting all the time (to the best of my understanding).

The Introduction refers to "genetic variability needed", but it is not so clear whether (natural) selection happens (or is the focus), or whether interest is in non-genetic transgenerational effects. Whether or not seed selection for the next generation was affected by natural selection should be discussed.

Related to the previous comment: natural selection for individual performance in mixture populations is expected to lead to an increase in competition (e.g. Griffing 1967 and later similar work on IGE). This seems to agree with the observation of taller plants containing more water. This could be discussed.

In the statistical analysis at the species level (L464-466) it seems a species effect is included as a random effect? That is surprising. Why?

Detailed comments

L21: My impression is that, in Figure 3b Fertilizer, x-axis values 2 and 4, these differences are not significant. IF so, is this statement warranted?

L24: Taller with more water seems to be indicative of more competition. This could be discussed (in Discussion).

Introduction: please partition the Introduction into several paragraphs. At present, it is hard to read. For example at L44, L50, L61, L69.

Can something be said about the selection history of the source material? Have all these varieties been selected for monocrop performance?

L81: It seems Figure 2b shows differences for facilitation, not (or sometimes) for competition, which contradicts this sentence. Maybe y-axis labels in Figure 2b and c have accidentally been swapped? Anyway, as suggested above, I suggest dropping the distinction between less competition and more facilitation. Hence, I suggest dropping panels b and c.

For mixtures, the meaning of "same coexistence history" is not fully clear. Is this 2 with a 2 history, or also e.g. 2 with a 4 history, etc. And how often?

Figure 2: I propose to explicitly state in the text that this (RII) is a measure relative to single plants.

L100: In the context of the current manuscript, the "corresponding monoculture" is not fully clear. Does it also mean: "with the same coexistence history" as the focal mixture?

L105-107: Is this sentence warranted given the absence of any indication of statistical significance (P = 0.67)?

Figure 3b: For easier interpretation: Could the y-axis be presented on the original (not square root transformed) scale? So can the estimates be back-transformed even though the P-values come from the transformed data?

Figure 4 panel e,f, monocrop (x = 1) seems to suggest that plants become more competitive when grown in the mixture (in the previous 2 generations), which would agree with the theory (Griffing 1967).

L456-470 The description of the statistical analysis is very verbal. A model presented as a mathematical equation would be much easier to read, and particularly to find back.

The Net Biodiversity Effect (L440) could be explained much easier, by first stating that it measures overyielding relative to monocrops, giving a simple equation for that, and only after that splitting this up into a main effect (the CE) and an interaction (the SE). Then the interpretation of the CE and SE should be clarified. (This is very similar to general and specific combining ability, an analogy that may help the reader).

L457: I may have missed it, but what is LER?

References

Griffing, B. (1967). Selection in reference to biological groups I. Individual and group selection applied to populations of unordered groups. Australian Journal of Biological Sciences, 20(1), 127-140.

---

## [Author Response]

Essential revisions:In your revision, please respond point-by-point to these essential revisions, which all reviewers agreed upon. In addition, the individual reviews, which are provided below, may contain some further points of interest.1) The major result as suggested in the title seems to be reduced competition of offspring with a co-occurrence history. However, reviewers feel that results about biodiversity and yield effects should be the focus: greater weight is given to results on RII, whereas results on NBE/yield are only marginally considered. The emphasis should rather be put on NBE and yield. They are the variables of interest to characterize the consequences of plant-plant interaction at the group level and they are the ones classically used in the BEF field (see Zuppinger-Dingley et al., Nature, 2014).

While we agree that NBE and total yield are the main measures classically used in the BEF and intercropping fields, we strongly believe that the added value of this project is precisely that we use a plant-plant interaction perspective (i.e. plant’s eye view) to improve our understanding in the BEF field. Rather than BEF relationships, we believe that plant-plant interactions are the core of this study, and this manuscript proposes to apply plant-plant interaction methods and metrics in an ecological approach to intercropping, also to understand mechanisms underlying potential NBE.

RII is a metric that has been extensively used in the field of plant-plant interactions and facilitation, such as Schöb Nat. Ecol. Evol. (2018), Diaz-Sierra Methods Ecol. Evol (2017), Schöb New Phyt (2014), Michalet Fun Ecol. (2013).

The classic method used in BEF studies, i.e. the additive partitioning of Loreau and Hector, compares the performance of mixtures vs monocultures. Because the reference is a monoculture, this method does not allow to understand the processes happening within monocultures, where plants can already suffer more or less from intraspecific competition. Since the core of this project was to study plant-plant interactions, we focused on the Relative Interaction Index because it does give additional information regarding the behaviour of plants both in monocultures and mixtures. For this metric, the reference is the individual plant growing in isolation, i.e. with zero interactions. This allows to compute and compare plant-plant interactions between different monocultures (in our case, fertilized vs unfertilized, or with different community histories). Furthermore, this index gives more insight into the mechanisms driving the classic net effects *sensu* L and H: indeed, if a plant grows more in a community compared to a single plant, this means that the net effect has at least a facilitative component. In contrast, if a plant grows less in a community in comparison to single plants, this means that the net effect has a competitive component.

Thus, we feel that it would be regretful to diminish the weight of RII and plant-plant interactions in the manuscript. However, we think it is reasonable to give an equal weight to community-level variables, such as NBE and yield, and we therefore developed these results more extensively in the discussion.

2) However, the significance of the effect of coexistence history on NBE in the fertilized treatment is not apparent in the manuscript. The analysis of fertilized vs. unfertilized treatments should be improved to better show if indeed there is an indication of greater net biodiversity effects under fertilized conditions. The analysis of traits should also be improved by separating between- from within-species variation and referring to work that shows different traits respond differently (height generally converging and leaf traits generally diverging, work by Roscher et al.).

The posthoc table of the significant interaction between fertilization and coexistence history was now added to the Supplement (Supplementary file 1d). The posthoc test does reveal that there is a marginally significant difference between coexistence history treatments in fertilized conditions, with a p-value of 0.0587.

The analysis of between and within species variation for traits has already been performed (see SI): we did the analyses at the plot level [i.e. between species] and at the species level [i.e. within species]. The very relevant work of Roscher at al has now been included and discussed in more detail. Thank you for drawing attention to it.

3) The discrepancy between 1) and 2) is very hard to understand and must be discussed by the authors. The way the single-plant reference is used for RII may affect the interpretation: is the apparent reduction in competition due to the mixture coexistence history, or due to the single reference plant? Because competition is always measured in terms of yield effects, the RII and total yield results would seem to rely on the same data. Hence, how can there be a true decrease in competition without an increase in yield? If this does not become clear, then the paper lacks a clear message.It is quite difficult to interpret RII and NBE as we do not know how the reference values were computed and if the same reference values were used across treatments. Because these indices are relative to single plants or monocultures, changes in index values can be caused by changes in the reference values (i.e. single plant or monoculture, respectively), not only by changes in the value measured in the mixtures.

RII is a measure of competition/facilitation and it indeed correlates well with CE from Loreau and Hector (in both fertilizing conditions) -> see Figure 5. Therefore, this is consistent, in the sense that higher RII (=reduced competition and/or increased facilitation) correlates with higher complementarity effects.

In theory, there is no reason why RII has to correlate with NE, as NE is driven both by CE and SE. In our case, RII actually correlates negatively with SE. Therefore, the positive correlation with CE is compensated by the negative one with SE, and in the end, we observe no correlation with NE. An SE can easily be explained by compensatory changes in RII of two species composing the mixture. If the high yielding species shows a 20% higher RII in mixture than in monoculture and the low yielding species shows a 20% lower RII in mixture than in monoculture, then we get a net RII change of 0, but a positive NE due to a positive SE.

Another reason that might explain the discrepancy between RII and NBE is the different levels of reference plants: indeed, RII uses a single plant as a reference, while NBE uses a monoculture. To further investigate this, we calculated RII in the mixtures using the individual in monoculture as reference hereafter called RII_monoculture:

RII_monoculture = (yield of individual of species A in the mix – yield of individual of species A in mono)/(yield of individual of species A in the mix + yield of individual of species A in mono). RII_monoculture significantly correlates with the complementarity effects from Hector and Loreau (see Author response image 1, p-value < 0.001). Furthermore, when running the linear model on this new index, we did not find any significant effect of coexistence history on RII_monoculture (see Author response table 1). This is in line with the lack of a significant history effect on CE and emphasizes the role of using single individual plants as a reference to better understand changes in plant-plant interactions with coexistence history that underly NBE.

**Author response table 1. sa2table1:** Anova table for RII_monoculture.

	*NumDF*	*DenDF*	*F value*	*Pr(>F)*
*Fertilizer*	1	7.883	1.306	0.286645
*History*	1	180.459	0.9898	0.321121
*Diversity*	1	15.098	0.1488	0.705034
*Fertilizer x history*	1	180.367	7.8531	0.005628 **
*Fertilizer x diversity*	1	175.703	0.217	0.641907
*History x diversity*	1	175.84	0.9712	0.325747
*Fertilizer x history x diversity*	1	175.641	0.2465	0.620166

**Author response image 1. sa2fig1:** Complementarity effects.

**Author response table 2. sa2table2:** Posthoc test for the significant interaction on fertilizer x history.

	estimate	SE	df	t.ratio	p.value
no diff – yes diff	0.129	0.0651	13.2	1.981	0.2436
no diff – no same	0.0964	0.0562	180.7	1.715	0.3187
no diff – yes same	0.059	0.0724	19.6	0.815	0.8466
yes diff – no same	-0.0327	0.0728	19.7	-0.449	0.9691
yes diff – yes same	-0.07	0.0546	177	-1.284	0.5745
no same – yes same	-0.0374	0.0794	27.1	-0.471	0.9649

Finally, the reference values of the single individuals per species were computed per fertilizer and coexistence history. And these reference values do indeed change with coexistence history (Figure 5 —figure supplement 1 and 2).

We chose this way of calculating these metrics as this allows to explicitly distinguish the effects of coexistence history on the interactions, independently of the baseline effect on plant performance overall. This follows the classic framework of plant-plant interaction work (see Michalet et al. 2014 Fun Ecol for instance). If we had kept the same reference plants for all coexistence histories (say single with single history, for instance), then we would not have been able to determine if the observed effect in the communities was due to the neighbour effect or because coexistence history was affecting plant performance in general (i.e. both single plants and plants in communities).

To investigate further the potential changes in reference plant performance, we calculated “RII coexistence” for each community (as they did in Michalet et al. 2014 for the effect of altitude on plant-plant interactions). This allows to see whether, within each diversity level (i.e. single, monocultures, mixtures), coexistence had an effect *per se* on plant performance in general.

This additional index allows to see that the effect of coexistence history on the performance of single plants varies across species: in the case of oat, flax and wheat, coexistence history does not change the performance of single individuals. In the case of camelina, coriander and lentil, the single plants grew worse when they had a community history than when they had a single history, which corroborates our results regarding adaptation in response to coexistence history. Indeed, the fact that single plants coming from single plants perform better than single plants coming from communities already suggests that plants have adapted to their surrounding community. Therefore, we believe that changes in these reference plants are already an effect of coexistence history, and that they should not be ignored by taking the same reference for all coexistence history treatments.

When calculating changes in plant performance in response to coexistence history within monocultures, we see again that this “RII monoculture” index is mostly negative, which means that the performance of plants with a mixture or single history is lower than plants with a monoculture history. This corroborates our adaptation results: in monocultures, plants coming from monocultures do better than plants coming from singles or mixtures. It also shows that coexistence history has an effect of plant performance *per se* and justifies the choice to take this effect into account when calculating the effect of neighbour on plant-plant interactions (i.e. by taking the appropriate reference).

This choice of index can explain why the effect of coexistence history on relative metrics (such as RII) does not appear in absolute metrics (such as total yield). We also suggest that the limited timeframe of this study – two generations – might be the reason for the lack of more significant changes in total yield. However, looking at the trend, we believe that there are clues indicating increased yield with a common coexistence history. We discuss this now extensively in the main manuscript (L211-226).

These additional results have now been added in the supplement (Figure 5 —figure supplement 1 and 2).

4) Compared with the SGH, there is as much evidence for the opposite response of increased biodiversity effects with improved environmental conditions. The authors should discuss their absence of evidence for SGH.

We believe that we actually see evidence for the SGH, according to our initial hypotheses: “we expected more facilitation and/or less competition in conditions of reduced soil fertility”. This is what we observed: in fertilized plots, competition was stronger and/or facilitation weaker than in unfertilized plots. Furthermore, in unfertilized plots only do we see “true” facilitation in mixtures (i.e. positive RII in mixtures and negative RII in monocultures). Furthermore, we suggest that increased biodiversity effects with improved environmental conditions are also in agreement with the SGH: if competition is more important or more intense in fertile conditions due to higher plant growth, then the benefits of niche differentiation coming from increasing diversity should also be higher. This has been visible in several intercropping studies, where authors have found increased biodiversity effects in highly-productive systems (see Li et al. 2020 Nat Plants, and Chen et al. 2021 Nat Plants). *(5)* The authors should tone done facilitation and focus on reduced competition (as they do in the title and abstract). They should probably omit the partitioning of the competition index into competition and facilitation components. For example, when RII goes from -0.5 to -0.3 using the same single isolated plant yield as a reference, it means that single plant yield has increased in the mixtures. This can be caused either by higher facilitation or lower competition. How can you tell the two mechanisms apart?

We removed the partitioning into competition and facilitation components as indeed a reduction in competition could also mean an increase in facilitation, and vice-versa. We also tried to rephrase and be more careful in our statements, such as “reduced competition and/or increased facilitation” to be more precise as it is indeed impossible to disentangle the two mechanisms when RII is negative in both the monoculture and the mixture (but less negative in the mixture). (6) Results for the second year might be included in the SI.

We collected only partial data after year one, as this was considered as an intermediate stage, where adaptation was less likely to give significant results. Notably, we put fewer efforts into collecting data at the individual-level and reduced the number of traits measured, which prevented us from having a full picture of the response of plant-plant interactions as well as of the trait space. Therefore, we decided to not present these partial pieces of data in the study.

7) Potential mechanisms underpinning transgenerational effects and reduced competition should be better explained. Thus, for the first it should be mentioned that it could occur by evolution via sorting out from standing variation (for highly selfing species if at least there was variation between initially sown genotypes), recombination (probably minor contribution in only 2 generations) or mutation (even less likely) or by epigenetic/physiological carry-over processes.

This was more extensively discussed in the discussion based on these helpful recommendations (L271-89).

8) The methods should be described in more detail.

This was improved according to the detailed comments below.

Reviewer #1 (Recommendations for the authors):I think the manuscript can be improved by removing the first index (RII) and focusing on plot-level analysis. This RII is not commonly used and not very explicit for the reader compared to NBE (or NE). Overall, the plot level data and trait data point to the same direction, which is no increase in complementarity between species over a generation of coexistence. I would focus on this message and discuss the limits of the design to detect a potential increase in complementarity if they were to occur as described in grasslands (inbred lines with low evolutionary potential, short evolutionary time, alternate rows which limit interactions, etc).

As indicated in the response to the review summary provided by the review editor (i.e. point (1) above), we feel that the plant’s eye view represented by the RII metric was the core of the study and of this manuscript; therefore, we think that it would be regretful to remove this element, which brings additional information regarding plant-plant interactions and how they relate to community-level metrics such as NE. However, we reconcile now extensively in the discussion the individual-level results and plot-level results.

I would avoid commenting on "trends" and non-significant results in the "Results and Discussion" section. I would also add some significance measures in this section (either p-value or stars on the Figures, or ANOVA Tables).

We added stars to the figures and removed all the comments regarding trends.

Line by line comment:l. 15: Why using seeds selected in monoculture could compromise yield benefit in mixtures? This is explained in the following sentence, but we lack the connection with this sentence.

This was rephrased.

l. 37: ref Vandermeer, J.H. (1992) The Ecology of Intercropping, Cambridge University Press.

This was added.

l. 44-47: ref Meilhac, J. et al. (2020) Both selection and plasticity drive niche differentiation in experimental grasslands. Nat. Plants 6, 28-33.

This was added, thank you.

l. 44: The first sentence is very general and does not convey any information. In the second sentence, it needs to be explained why the use of commercial seeds bred for monocultures might not be optimal to promote positive diversity effects.

This was made clearer.

l. 52: "changed and evolved" is redundant.

We kept only “changed”.

l. 54: I would provide more information on the species (species or functional groups) and say if they are commonly grown as intercrops.

This was added.

l. 56: mesocosms is only used here and not defined.

This was defined.

l. 57-8: "We selected open-pollinated varieties", This does not seem to be the case for oat, wheat, and lentil in the Methods section.

As we could not know the initial amount of standing variation, we clarified that we selected, whenever possible, open-pollinated varieties.

l. 62: It is not clear what is the difference between this Relative Interaction Index, and the classical Relative Yield Total used in the partitioning of Loreau and Hector at this stage. Why use both? What is the information added by Relative Interaction Index?

The classic method of Loreau and Hector, and also the Relative Yield Total or Land Equivalent Ratio, compares the performance of mixtures vs monocultures. Because the reference is a monoculture, this method does not allow to understand the processes happening within monocultures, where plants can already suffer more or less from intraspecific competition. Since the core of this project was to study plant-plant interactions, we focused on the Relative Interaction Index because it does give additional information regarding the behaviour of plants both in monocultures and mixtures. For this metric, the reference is the individual plant growing in isolation, i.e. with zero interaction. This allows to compute and compare plant-plant interactions between different monocultures (in our case, fertilizer vs unfertilized, or with different community history). Furthermore, this index gives more insight into the mechanisms driving the classic net effects *sensu* L and H: indeed, if a plant grows more in a community compared to a single plant, this means that the net effect has at least a facilitative component. On the contrary, if a plant grows less in a community in comparison to single plants, this means that the net effect has a competitive component.

l. 69 to 79: it should be stated that this hypothesis applies to mixture communities only.

This was specified.

l. 77-8: The SGH arrives a bit "out of the blue" here. It needs to be defined and developed within the context of intercropping before.

This was added in the intro (L68-78).

l. 81-82: Figure 2 and Table S1 suggest that the "history" treatment effect is primarily driven by facilitation, not by competition. Put differently, the difference between "same" and "different" evolutionary history is much bigger for the positive interaction index than for the negative ones. The sentences here suggest the opposite, i.e. greater effect on competition than on facilitation.

This was changed as indeed, the distinction between increased facilitation/reduced competition is blurry. We therefore tried to give equal weights to facilitation and competition.

l. 83 and throughout the manuscript: "SI" can be removed before supplementary material references.

This was removed.

l. 91: "less competition": I think it should be "more facilitation" instead (cf previous comment).

We changed to less competition and/or more facilitation to be consistent with the general comments.

l.99 to 105: this whole part only comments on non-significant results.

The interaction effect between fertilization and coexistence history on net biodiversity effect is significant. We now added the full table of the corresponding posthoc test to the supplement and referred to it in the main text. While the p-values of this posthoc test are indeed higher than 0.05 (p-value = 0.0587 for the history effect in fertilised conditions), this is nonetheless a marginally significant result and is worth reporting, especially considering the natural variation due to the external conditions and inherent environmental variability. We also looked into fertilized and unfertilized plots separately. We did find a significant effect of coexistence history in the fertilized plots (p-value of 0.02364, see Author response table 3); however, separating all the analyses would be confusing for the reader. We therefore suggest to keep the original model and the marginally significant effect revealed by the posthoc test (Supplementary file 1d).

**Author response table 3. sa2table3:** Author response table.

	Sum Sq Mean Sq NumDF	DenDF F	value	Pr(>F)	
history	2801.42 2801.42 1	81.814	5.3175	0.02364	*
diversity	580.76 580.76 1	14.804	1.1024	0.31059	
history:diversity	41.19 41.19 1	80.246	0.0782	0.78048	

l. 112: I could not find to which data and test the p-value and F value reported here refer.

This was clarified and the corresponding tables for the ANOVA and the posthoc test were referenced and fully added to the Supplement. l. 142-144: "height community-weighted mean was not significantly different between coexistence histories".

This was changed. We also added precisions regarding the significant effect of coexistence history on height but at the species-level (not at the community level). Paragraph 120-145: I would add the results on Functional hypervolume and PAR here. I think it is especially interesting to discuss the differences in PAR interception between communities with different coexistence histories (you do have significant evidence of complementary effects increased in mixtures with a mixture history here).

We moved the PAR results to the main text but decided to keep the functional diversity as a supplement, as there was no response of history for this measure.

l. 172-173: "In fertilized conditions, this shift in plant-plant interactions was associated with an increase in overyielding": I could not find the statistics supporting that statement. In Table S4, the Fertilizer x history effect is significant, but we do not have the tests of the history effect within each fertilization treatment. Also, the p-value used to support this in the result section (l. 103) is higher than 0.05.

The posthoc test was now added in the supplement (Supplementary file 1d), showing the marginally significant effect of coexistence history in fertilised plots (p-value = 0.0587). As mentioned above, we also looked into fertilized and unfertilized plots separately. We did find a significant effect of coexistence history in the fertilized plots (p-value of 0.02364, see Author response table 4); however, we feel like separating now all the analyses would be confusing for the reader and suggest to keep the original model and the marginally significant effect- shown through the posthoc test.

**Author response table 4. sa2table4:** Author response table.

	Sum Sq Mean Sq NumDF	DenDF F	value	Pr(>F)	
history	2801.42 2801.42 1	81.814	5.3175	0.02364	*
diversity	580.76 580.76 1	14.804	1.1024	0.31059	
history:diversity	41.19 41.19 1	80.246	0.0782	0.78048	

l. 194-200: this discussion should be confronted with the Stress-Gradient Hypothesis, as presented in the Introduction. Indeed, the SGH predicts the opposite pattern as the one observed in the study, i.e., stronger biodiversity effects in low-input systems.

The SGH predicts stronger facilitation in low-input systems, but not necessarily stronger biodiversity effects. Indeed, if competition is lower in low-input systems, the benefits of niche differentiation (so reduced competition) would also be lower. In highly productive systems where plants strongly compete, the benefits of niche differentiation would then be bigger. Of course, facilitation would be lower in highly-productive systems, but here in our Swiss agroecosystem, we suggest that competition is the dominant interaction, even in unfertilized conditions, as the soils were initially quite fertile. This result was found in other intercropping studies, including a large meta-analysis (see Li et al. 2020 in Nat Plants), where they found that biodiversity effects were higher in highly-productive systems.

l. 205-206: "a reduction in trait variation favoured increased yield benefits in mixtures": I cannot find any result supporting that statement.

This was rephrased to focus on the effect of coexistence history on traits.

l. 209: "plants might have adapted to express the phenotype that would maximise their fitness": again, I do not see any relationship between traits and fitness in the results.

This was rephrased to focus on the effect of coexistence history on traits.

l. 218-222: this result should be presented in the main text.

This was moved to the main text.

l. 227-228: "we specifically selected open-pollinated varieties in order to ensure a minimum amount of genetic variability". There are two problems here. First, it was not open-pollinated varieties for all species. Second open-pollinated varieties have more genetic variability than other varietal types such as inbred lines or hybrids.

This was clarified and we stated that open-pollinated varieties included camelina and coriander.

l. 262: 30 cm of what?

This was clarified.

l. 268: were the metal frames only belowground or also aboveground?

They were only belowground (30 cm belowground). This was specified.

l. 286 and the whole sub-section: It is not easy to assess the extent of genetic variation within each species in this section. It seems that there was no within-species variation for wheat, oat, and lentil, and maybe some variation for the 3 others. I think having this kind of information would be important to interpret the results phenotypic changes can occur both through selection and plasticity, depending on the amount of standing variation in the species.

We did not assess the initial standing variation within our crop populations, as investigating the underlying mechanisms was not the main point of this study and the genetic analyses would have required much more time and funding. We described the seed sources in the Method part as precisely as possible considering the information available, but regarding the mechanisms, we can only speculate. It is however true that the most common species (i.e. wheat and oat) probably had the lowest standing variation, as these two species have been selected and controlled for homogeneity for decades. This was added as an element of discussion.

l. 335 to 337: I first did not understand that single plant plots were grouped together in separate beds. In fact, I only got it later with the picture in Figure S10. I would reword this part.

This was rephrased, and we also moved Figure S10 to Figure 1 to give readers a better grasp.

l. 340-342: We have no information on how the different species were arranged in 4-species mixtures.

This was specified.

l. 342-343: Are these densities the monoculture densities? If yes, how were these densities adjusted in mixtures?

Yes indeed, these are the monoculture densities. The densities were not adjusted in the mixtures (i.e. if we had 25 individuals per vertical line in the monocultures, we kept 25 individuals per line in the mixtures for this species) to avoid concomitant effects of density changes on productivity. This was specified in the text.

l. 400: "yield": is it grain or biomass yield?

Yield is defined as grain yield.

Also, how was "Yield single" computed? Did you consider only a single plant from the same year, same fertilization, with a pure single plant history? Were the different single plants from the same species averaged, corrected for design effects, etc?

This was specified in the Methods and discussed in the response to the general comments. We always used the average value over the four replicates from the same year, with the same fertilization and coexistence history combination, e.g. a single plant with a monoculture history in unfertilised plots as a reference for monoculture history treatments in unfertilised plots.

l. 410: "proportional": amend to "relative".

This was changed.

l. 418-420: Why not use directly the Loreau and Hector partitioning then?

As mentioned earlier, the RII metric allows to have a finer picture of plant-plant interactions, both at the individual and also at the community level. This allows to compare different monocultures but also to disentangle, in some cases, net facilitative from competitive effects. Since the core of this study was on changes in plant-plant interactions underlying mixture benefits, we feel that this is the right metric. Furthermore, it has already been used in several facilitation and competition studies, such as Schöb et al. Nat. Ecol. Evol. (2018), Diaz-Sierra et al. Methods Ecol. Evol (2017), Schöb et al. New Phyt (2014), Michalet et al. Fun Ecol. (2013).

l. 431 to 436: Since RIINet and NInC are highly correlated, it is not surprising that you get the same results: you just repeated your analysis with a redundant index. I would either remove or just mentioned that RIInet and NIntC were strongly correlated and thus provided identical results.

We removed this extra index as the information is indeed redundant.

l. 440: as for RII and single plant yield, how were monoculture yields computed here? Was it averaged across replicates of the same species, only considering replicates with a "pure" monoculture history?

This was specified in the methods; it was averaged across replicates of the same species, fertilizing condition and coexistence history.

l. 451-452: Community-Weighted Mean needs to be defined and referenced.

This was added.

l. 456 and 457: "NIntC" should be amended to "RII" and "LER" should be amended to "SE"

Thank you, this was corrected.

Table S17: Which PAR values were used in this analysis of variance? Was it a single date, or averaged across dates?

We took all the dates (indicated in L541) but added day of year as a random factor in the model, to have the response “across date”.

Figure S10: I would move this Figure to the main text or provide an illustration of the experimental set-up there to help the readers visualize what are beds, plots, and how are the different species arranged, notably in mixture plots.

This was moved to the main text along with Figure 1.

Reviewer #2 (Recommendations for the authors):1. Please give a more balanced discussion of the proposed mechanisms for how co-occurrence can lead to more facilitation. I see how co-occurrence could lead to trait displacement and less niche overlap, so less competition. But what is the facilitation part of this? Please include something in the intro that covers this.

We included a paragraph covering the evolution of facilitation in the introduction (L49-57).

2. Your RII_fac and RII_comp appear to be superficially separated based on whether the net effects of neighbors were positive or negative, is this correct? What would happen if one species in a mixture plot had net positive effects of neighbors and another species had net negative effects of neighbors? Would you calculate the whole plot as just the sum of that? Or how is it calculated in that case?

The distinction between facilitation and competition was dropped as indeed it was redundant and did not bring any more insight than the net RII.

3. Also, how do you separate intraspecific competition and interspecific competition from these metrics? For example, if intraspecific comp > interspecific comp, you would still see a less negative RII_comp at higher diversity. But this isn't necessarily because competition is actually weaker at higher diversity (but instead a shift from one type of competition to another). Similarly, if intraspecific comp > interspecific comp, you could also see these shifts in RII_fac. A stronger RII_fac value could be completely driven by just alleviation of intraspecific competition at higher diversity.

We think that stronger intraspecific vs interspecific competition is an inherent mechanism underlying positive BEF effects. This was actually one of the reasons why we focused on the plant’s eye view in BEF studies and quantified plant-plant interaction metrics here. And we agree with your assessment of changes in RII_fac from monoculture to mixture being potentially due to alleviation of intraspecific competition – but this only if there is also facilitation going on. We would not get RII_fac (i.e. individuals growing bigger in a community than as single plant) when there is no facilitation but only alleviation of intraspecific competition. In any case, we decided to remove the distinction between RII_fac and RII_comp, as outlined above.

4. I'm thus consistently tripped up by the many phrases in the manuscript where you say things like: "shifted towards weaker competition, and in some cases, stronger facilitation" (lines 81-82). All of your RII values are net effects, so how can you conclude that what drives the changes you see is due to competition or facilitation? Why not: "shifted towards weaker competition and/or stronger facilitation (though teasing out the differences is not possible in our given dataset)".

This was changed throughout the manuscript, and we tried to be more precise and careful in our statements.

5. Overall trends I see in RII: most things do worse when growing next to same species neighbors (vs. alone), but the same or better if they grow next to heterospecific neighbors. This gets stronger if they have coexisted for multiple generations. And the positive aspects of it are most strongly affected by coexistence. Your data indicate that this ISN'T due to trait divergence which is super interesting! You also show a strong correlation with selection effects which indicates that a couple of species are driving the patterns. Did you take a look at how selection effects were correlated with RII? Do you know which species are doing that? Have you split into plots with and without legumes? Do legume effects on neighbors get stronger with coexistence history? The only other mechanism I could imagine is more specialization of enemies over time leading to stronger dilution effects? Please discuss these possible mechanisms in some way.

Since all our 4-species mixtures necessarily included a legume, the plots without legumes were only monocultures and 2-species mixtures, and therefore it did not give a representative picture of the experimental design. We can only compare 2-species mixtures if we want to compare with and without legumes, therefore we lose many plots and, as we saw in the other results, planted diversity is often significant. This is notably the case for RII and CE, which are both higher in 4-species mixtures compared to 2-species mixtures and where we expected the most important effects of coexistence history. When only considering the 2-species mixtures and separating with and without legumes, we for instance see a positive effect of coexistence history on RII in fertilized plots with legumes, but when looking at CE, we see a positive effect of coexistence history in fertilized plots without legumes. Disentangling the role of legumes would require having 4-species mixtures without legumes and much more in depth analyses that would fall out of the scope of this paper, which aimed at investigating general patterns across species. Furthermore, since the plots were reshuffled each year to avoid soil legacy effect, dilution effects of pathogens are, in our opinion, unlikely. However, this is highly speculative; further research and analyses are certainly needed to investigate the mechanisms behind these results.

6. The loss of variability in each species when they have coexisted for multiple generations also seems to point towards some kind of restricted gene flow. How much interbreeding was there between plots? Is there something about the environment that you could imagine selecting for these very tall plants? Also, the fact that they are getting more like each other overall, but still having more positive effects on each other is very strange. Need more context about how this might be explainable.

We could not measure precisely how much interbreeding there was between plots; however, since the plots were very close to each other, we believe that there was no limitation to interbreeding between plots. The selection for tall plants has been referenced and observed previously, notably because competition for light is asymmetric (see Lipowsky 2015, Falster 2003), meaning that a taller plant gets a much larger amount of the resource than a shorter one. This puts a high selection pressure on species to evolve plasticity for increased plant height in response to lower light. We were also surprised by the evidence pointing towards character convergence; however, we do see an increase in light interception with a common coexistence history, which points towards improved resource use and increased niche differentiation, at least for light. We therefore believe that the measured traits did not allow us to see the changes in resource use; light interception can be influenced by leaf architecture or vertical leaf distribution, which we did not measure here.

Reviewer #3 (Recommendations for the authors):Background on main comments / further general commentsThe presence of clear effects of the coexistence history on competition together with absence of effects on yield is worrying. Particularly because effects on competition relative to single plants (Figure 2) are statistically very significant while those on Net Effect relative to monocrop are borderline (P = 0.0715). At the same time, the plot of total yield excludes the results for single plants (these should be included).

The results for single plants were included in Figure 3b.

The decrease in competition due to the same coexistence history shown in Figure 2 is relative to single plants. Hence, the comparison is relative. Do we really see a decrease in competition here, or is the single plant getting worse due to the coexistence history? Importantly: It is unclear in Figure 2a whether the two points that make up a pair of points for the same x-axis value are relative to a single plant with the same coexistence history. For example, for x = 1 (monoculture) the two y-axis values are about -0.3 and -0.2. Is the reference single plant the same for these two points the same, so that we can also infer that the difference between both points is 0.1? Or is the reference single plant for the first point (-0.3) a single plant with also a Different co-existence history, and the reference single plant for the second point (-0.2) a single plant with also a Same co-existence history. If the latter is the case, then both points have a different reference, and the apparent difference of 0.1 may very well originate from having two different single plant references, rather than an effect on competition in the monocrop.

See response to main comments, in particular (3).

Related to the previous comment: I cannot reconcile the results for No Fertilizer in Figures 3a and 3b. NE is a measure of overyielding relative to monocrops. Figure 3a NoFertilizer shows that overyielding is greater with Different. However, 3b shows greater total yield with Same. This strongly suggests that the two points making up a pair in Figure 3a do not have the same monocrop as the reference. Hence, then we don't know whether the effect is due to the coexistence history of the 2-mixture or due to the co-existence history of the monocrop used as reference. Things get very confusing because the reference point seems to be shifting all the time (to the best of my understanding).

See response to main comments, in particular (3).

The Introduction refers to "genetic variability needed", but it is not so clear whether (natural) selection happens (or is the focus), or whether interest is in non-genetic transgenerational effects. Whether or not seed selection for the next generation was affected by natural selection should be discussed.

We recognize that the underlying mechanisms remain unclear, as investigating these mechanisms was not the goal of the study. Therefore, we can only speculate regarding the potential mechanisms, and we have now added an extensive paragraph discussing the possibilities in the discussion (L271-289).

Related to the previous comment: natural selection for individual performance in mixture populations is expected to lead to an increase in competition (e.g. Griffing 1967 and later similar work on IGE). This seems to agree with the observation of taller plants containing more water. This could be discussed.

Height indeed generally converges towards taller plants in denser and more diverse communities; this was now more discussed in the discussion. Previous work shows the same trend, due to the asymmetrical character of competition for light (see Lipowsky et al. 2015). However, lower leaf dry matter content indicates less resource-conservative strategies and it has recently been associated with lower parental or ambient competition, so this trait goes in line with our interaction results (see Reich et al. 2014, Puy et al. 2020).

In the statistical analysis at the species level (L464-466) it seems a species effect is included as a random effect? That is surprising. Why?

For the analyses at the species level only, we added species as a random effect, because we wanted to see whether there was a response across all species and were not interested in the response of species *per se*.

Detailed commentsL21: My impression is that, in Figure 3b Fertilizer, x-axis values 2 and 4, these differences are not significant. IF so, is this statement warranted?

Indeed, for total yield the difference is not significant, but for yield benefits (=overyielding, or net effects), it is in fertilized plots. We clarified this.

L24: Taller with more water seems to be indicative of more competition. This could be discussed (in Discussion).

See response above.

Introduction: please partition the Introduction into several paragraphs. At present, it is hard to read. For example at L44, L50, L61, L69.

This was done.

Can something be said about the selection history of the source material? Have all these varieties been selected for monocrop performance?

These varieties come from standard seed providers used for agricultural purposes, hence they are usually used as monocultures. However, some less “usual” crops, such as camelina or coriander, surely have a shorter breeding history as wheat or oat.

L81: It seems Figure 2b shows differences for facilitation, not (or sometimes) for competition, which contradicts this sentence. Maybe y-axis labels in Figure 2b and c have accidentally been swapped? Anyway, as suggested above, I suggest dropping the distinction between less competition and more facilitation. Hence, I suggest dropping panels b and c.

We removed the partitioning into competition and facilitation components.

For mixtures, the meaning of "same coexistence history" is not fully clear. Is this 2 with a 2 history, or also e.g. 2 with a 4 history, etc. And how often?

This was clarified in the Figure 2 legend.

Figure 2: I propose to explicitly state in the text that this (RII) is a measure relative to single plants.

This was made clearer in Figure 2.

L100: In the context of the current manuscript, the "corresponding monoculture" is not fully clear. Does it also mean: "with the same coexistence history" as the focal mixture?

Yes, this is what it means. This was clarified.

L105-107: Is this sentence warranted given the absence of any indication of statistical significance (P = 0.67)?

This sentence was actually referring to the response of net effect (overyielding), but we moved the sentence earlier to be clearer and removed the commentary on the nonsignificant response of yield.

Figure 3b: For easier interpretation: Could the y-axis be presented on the original (not square root transformed) scale? So can the estimates be back-transformed even though the P-values come from the transformed data?

This was changed.

Figure 4 panel e,f, monocrop (x = 1) seems to suggest that plants become more competitive when grown in the mixture (in the previous 2 generations), which would agree with the theory (Griffing 1967).

Taller plants would indeed suggest more competitive plants. However, the plant interaction index does not indicate whether a plant is competitive or not, but rather whether they experience competition (i.e. whether their growth is more or less hindered by other plants). In that sense, lower LDMC indicates softer leaves, and consequently less resource-conservative strategy. On the contrary, a higher LDMC would increase a plant’s ability to cope with stress. We therefore believe that plants in the same coexistence history might be less “stressed” because they might experience less competition (i.e. this does not mean that they are not more competitive, but that at the community-level they experience less competition from their neighbours).

L456-470 The description of the statistical analysis is very verbal. A model presented as a mathematical equation would be much easier to read, and particularly to find back.

We added the general equation in the method for more clarity.

The Net Biodiversity Effect (L440) could be explained much easier, by first stating that it measures overyielding relative to monocrops, giving a simple equation for that, and only after that splitting this up into a main effect (the CE) and an interaction (the SE).

This was done.

Then the interpretation of the CE and SE should be clarified. (This is very similar to general and specific combining ability, an analogy that may help the reader).

The interpretation was added and clarified in the methods (L509-513).

L457: I may have missed it, but what is LER?

This was a mistake, thank you for pointing this out.